# Mercury deposition and redox transformation processes in peatland constrained by mercury stable isotopes

Chuxian Li [1] ✉, Martin Jiskra [2], Mats B. Nilsson [1], Stefan Osterwalder [3], Wei Zhu[1], Dmitri Mauquoy [4], Ulf Skyllberg [1], Maxime Enrico [5], Haijun Peng [1], Yu Song [1], Erik Björn [6] & Kevin Bishop [7]

Peatland vegetation takes up mercury (Hg) from the atmosphere, typically contributing to net production and export of neurotoxic methyl-Hg to downstream ecosystems. Chemical reduction processes can slow down methyl-Hg production by releasing Hg from peat back to the atmosphere. The extent of these processes remains, however, unclear. Here we present results from a comprehensive study covering concentrations and isotopic signatures of Hg in an open boreal peatland system to identify post-depositional Hg redox transformation processes. Isotope mass balances suggest photoreduction of $Hg^{II}$ is the predominant process by which 30% of annually deposited Hg is emitted back to the atmosphere. Isotopic analyses indicate that above the water table, dark abiotic oxidation decreases peat soil gaseous $Hg^0$ concentrations. Below the water table, supersaturation of gaseous Hg is likely created more by direct photoreduction of rainfall rather than by reduction and release of Hg from the peat soil. Identification and quantification of these light-driven and dark redox processes advance our understanding of the fate of Hg in peatlands, including the potential for mobilization and methylation of $Hg^{II}$.

The peatlands covering 3% of the Earth's land surface are hotspots for the production of neurotoxic methyl-mercury (methyl-Hg)[1]. This methyl-Hg can be exported to downstream aquatic systems and subsequently biomagnifies in the food web[2]. Peatlands receive atmospheric Hg largely through vegetation uptake of gaseous elemental Hg ($Hg^0$)[3,4], the dominant form of Hg in the atmosphere[5], including the possibility of surface adsorption of $Hg^0$. $Hg^0$ taken up by vegetation is further oxidized to divalent reactive Hg ($Hg^{II}$) via enzymatic reactions or by the action of reactive oxygen species[6]. Rainfall input of $Hg^{II}$ to peatlands also contributes, but it is smaller in magnitude (e.g., 20 – 30% of total Hg[3]). $Hg^{II}$ from both rainfall and oxidation of $Hg^0$ associated with plant uptake quickly bind to the thiol groups of natural

organic matter (NOM)[7], potentially forming immobile nanoparticulate $\beta$-HgS[8]. The Hg stored in boreal and subarctic peatlands is globally significant given that peatlands comprise 15 – 30% of terrestrial organic carbon[9,10]. Reduction processes may transform some of the deposited $Hg^{II}$ to volatile $Hg^0$, which can move vertically within the pore system of the peatland[11]. Net $Hg^0$ evasion from an open (tree-less) boreal peatland was measured with a micrometeorological method over the course of one year[12]. Variability in $Hg^0$ evasion rates along a thawing permafrost fen-palsa-bog gradient in the subarctic were related to different amounts of Hg stored in the peat[13]. While reduction and subsequent evasion might reduce the $Hg^{II}$ available for methylation, it also raises questions about peatlands as a long-term Hg sink[14], and the

[1]Department of Forest Ecology and Management, Swedish University of Agricultural Sciences, Umeå, Sweden. [2]Environmental Geosciences, University of Basel, Basel, Switzerland. [3]Institute of Agricultural Sciences, ETH Zurich, Zurich, Switzerland. [4]School Geosciences, University of Aberdeen, Scotland, UK. [5]Universite de Pau et des Pays de l'Adour, E2S UPPA, CNRS, TotalEnergies, LFCR, IPREM, Pau, France. [6]Department of Chemistry, Umeå University, Umeå, Sweden. [7]Department of Aquatic Sciences and Assessment, Swedish University of Agricultural Sciences, Uppsala, Sweden. ✉ e-mail: chuxian.li@slu.se

suitability of peatlands as archives of earlier Hg deposition[15,16]. Given the importance of peatlands in global, regional and local Hg cycles, it is crucial to understand Hg deposition and biogeochemical transformation processes in the peat. The post-depositional processes of Hg related to Hg reduction and oxidation could be resolved by analyzing the abundance and composition of Hg isotopes.

Hg stable isotopes enable us to constrain sources of Hg and its transformation processes because isotopes undergo fractionation during biogeochemical cycling (e.g., reduction and oxidation processes[17,18]). The fractionation either depends on mass (MDF, represented by $\delta^{202}Hg$) or is independent of the isotopic mass (MIF, represented by $\Delta^{199}Hg$ and $\Delta^{201}Hg$). Fig. 1 presents an overview of potential Hg MDF and Hg MIF of odd mass isotopes in peat soil systems. Plant uptake of $Hg^0$ favors light isotopes (i.e., lower $\delta^{202}Hg$, (-) MDF[19]), while direct rainfall supply of $Hg^{II}$ is not found to cause MDF and MIF. Peat $Hg^{II}$ is transformed to gaseous $Hg^0$ through photochemical, biotic and abiotic reduction processes. These processes leave residual $Hg^{II}$ enriched in heavier isotopes after losses of $Hg^0$ ((+) MDF[20–22]). At the peat surface, UV radiation in sunlight can reduce newly deposited Hg, either when adsorbed onto leaf surfaces[23] or when stored in leaf interiors[24]. Depending on the strength of the $Hg^{II}$ bonds to ligands[25], photo-reduction of $Hg^{II}$ can produce $Hg^0$ with either negative odd-mass MIF (ligands with N/O-Hg bonds[26]) or positive odd-

mass MIF (ligands with S-Hg bonds[24] prevalent in peat[27,28]) due to magnetic isotopic effects. In sub-surface peat soils, dark abiotic or biotic reduction of $Hg^{II}$ can occur[29]. Biotic reduction of Hg results in MDF without any significant MIF[30], similar to microbial methylation and demethylation of Hg with only significant MDF[17]. Dark abiotic reduction, as controlled by Nuclear Volume Effect (NVE), results in positive odd-mass MIF in product $Hg^{0\ 31}$. Jiskra et al.[14] estimated a 27% loss of Hg in boreal peat soil (a riparian zone soil with tree cover) via dark reduction by NOM, and Yuan et al.[32] reported a larger relative Hg loss in a forest ecosystem caused by NOM dark reduction (two thirds) than by microbial reduction (one third). In contrast, dark abiotic oxidation of $Hg^0$ to $Hg^{II}$ with positive odd-mass MIF was found based on experimental work with thiol compounds and humic acids[18]. This was explained by equilibrium fractionation[33]. Field studies have further demonstrated the quantitative importance of Hg dark abiotic oxidation in arctic tundra soils[34]. Despite these recent advances in the understanding of how Hg isotopes are fractionated by transformation processes, it still remains unclear how post-depositional processes are affecting the fate of Hg in peatlands.

In this study, we identify redox-related Hg post-depositional processes and associated magnitudes in an open (tree-less) boreal peatland system, Degerö Stormyr (64°11′N, 19°33′E, Supplementary Fig. S1). We have comprehensively investigated the uppermost meter of $^{14}C$ dated peat soil by combining measurements of Hg concentrations with the natural abundance of Hg stable isotopes in key compartments of the peatland ecosystems (atmosphere, peat soil, groundwater and soil gas). These investigations are made in two distinct peat microforms, slightly elevated (20 – 30 cm) hummocks and flatter lawns, which differ in the water table level relative to the peat surface and their characteristic vegetation composition. Both of these differences have the potential to affect Hg deposition rates and redox-related mobility processes. This study includes the reports of Hg concentration and isotopes in the peat soil gas of the unsaturated zone above the water table, as well as the Hg isotopes in dissolved gaseous Hg (DGM) of peat groundwater just below the water table.

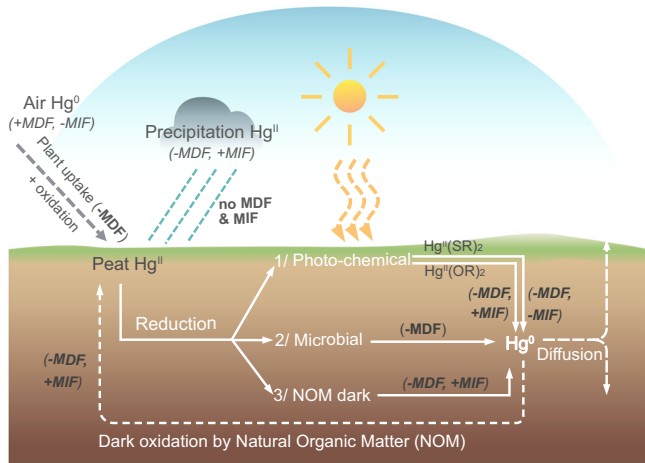

**Fig. 1 | Overview of potential redox-related Hg stable isotope fractionation in peatlands.** Hg stable isotope fractionation is either dependent on atomic mass (MDF) or independent of that mass (MIF, referred to here as odd mass since it has only been significantly identified in atoms with odd atomic mass) as related to potential biogeochemical processes in peat soils. Atmospheric $Hg^0$ isotopic signatures are characterized by (+)MDF and (-)MIF[95]. The $Hg^0$ in the high altitude atmosphere can be photo-oxidized to gaseous, water-soluble and particulate-bound $Hg^{II}$ with (-)MDF and (+)MIF, respectively[96]. These signatures embed in the rainfall. Plant uptake of $Hg^0$ controls deposition onto peatlands with a preference for light isotopes ((-)MDF[3,19]). Precipitation also supplies $Hg^{II}$ with non-significant isotopic fractionation expected when falling on the peatland. Once deposited to the peat, Hg can be reduced by i) photochemistry, ii) microbial activity, and iii) Natural Organic Matter (NOM) in the dark. Photoreduction of solid peat Hg can lead to $Hg^0$ with (-)MDF and (+)MIF when Hg is bound to sulfur ligands ($Hg^{II}(SR)_2$[20,24]), or (-)MDF and (-)MIF when bound to oxygen or nitrogen ligands ($Hg^{II}(O/NR)_2$[26]) due to magnetic isotopic effects. Microbial reduction of Hg only produces (-)MDF in $Hg^0$ without any significant MIF[30]. NOM-driven dark reduction can lead to $Hg^0$ with (-) MDF and (+)MIF, similar to photo-reduction on $Hg^{II}(SR)_2$, but as a result of the Nuclear Volume Effect (NVE[31]). The $Hg^0$ produced by reduction can diffuse upward to the atmosphere or downward into the deeper peat soils. NOM dark oxidation of $Hg^0$ to $Hg^{II}$ can also occur, resulting in $Hg^{II}$ with (+)MDF and (-)MIF due to equilibrium fractionation[18,33,34].

## Results and discussion
### Peat Hg accumulation rates and potential influences
Peatlands receive Hg mostly from atmospheric deposition through plant uptake of $Hg^0$ and rainfall $Hg^{II}$ supply. Post-depositional processes potentially result in a loss of $Hg^0$ back to the atmosphere as a consequence of biotic and/or abiotic reduction of $Hg^{II}$ to $Hg^0$. Both deposition and post-depositional losses of Hg are reflected in the measured peat Hg accumulation rates (AR). Modern Degerö hummock HgAR at 2000−2020CE is 14.4 ± 4.8 µg m$^{-2}$ yr$^{-1}$ (Fig. 2a). This is similar to those in two Southern Swedish hummock sites (17 µg m$^{-2}$ yr$^{-1}$ in Dumme Mosse for the period 1990−1995 CE[35] and 18 µg m$^{-2}$ yr$^{-1}$ in Store Mosse for the period 1990−2020 CE[36]). This HgAR is, however, more than twice as high as in the Degerö lawn for the same period (6.6 ± 1.6 µg m$^{-2}$ yr$^{-1}$, Fig. 2b; $P$ < 0.001, two-tailed T test). Such a significant difference between hummock and lawn sites just 5 m apart on an open peatland can be explained by either a greater Hg sequestration in hummock or a higher post-depositional Hg loss from the lawn. Enhanced Hg sequestration in µg m$^{-2}$ yr$^{-1}$ can be achieved by higher peat Hg assimilation in ng g$^{-1}$ and/or higher peat biomass production (g cm$^{-2}$ yr$^{-1}$, Fig. 2a–d; Supplementary Text S1; Table S1; Figs. S2 and S3). These two factors have seldom been explicitly discussed together in details of previous studies[3,37,38].

Surface living vegetation on hummocks has an average Hg concentration of 29 ± 5.5 ng g$^{-1}$, which is greater than that of the vegetation on lawns (21 ± 1.9 ng g$^{-1}$, 1σ, $n$ = 3, top 3 cm based on the length of the green section of peat moss, Supplementary Table S2). In the hummock profile, *Sphagnum fuscum* is dominant whilst *Sphagnum* section *Cuspidata* (including *Sphagnum balticum* and *Sphagnum recurvum*

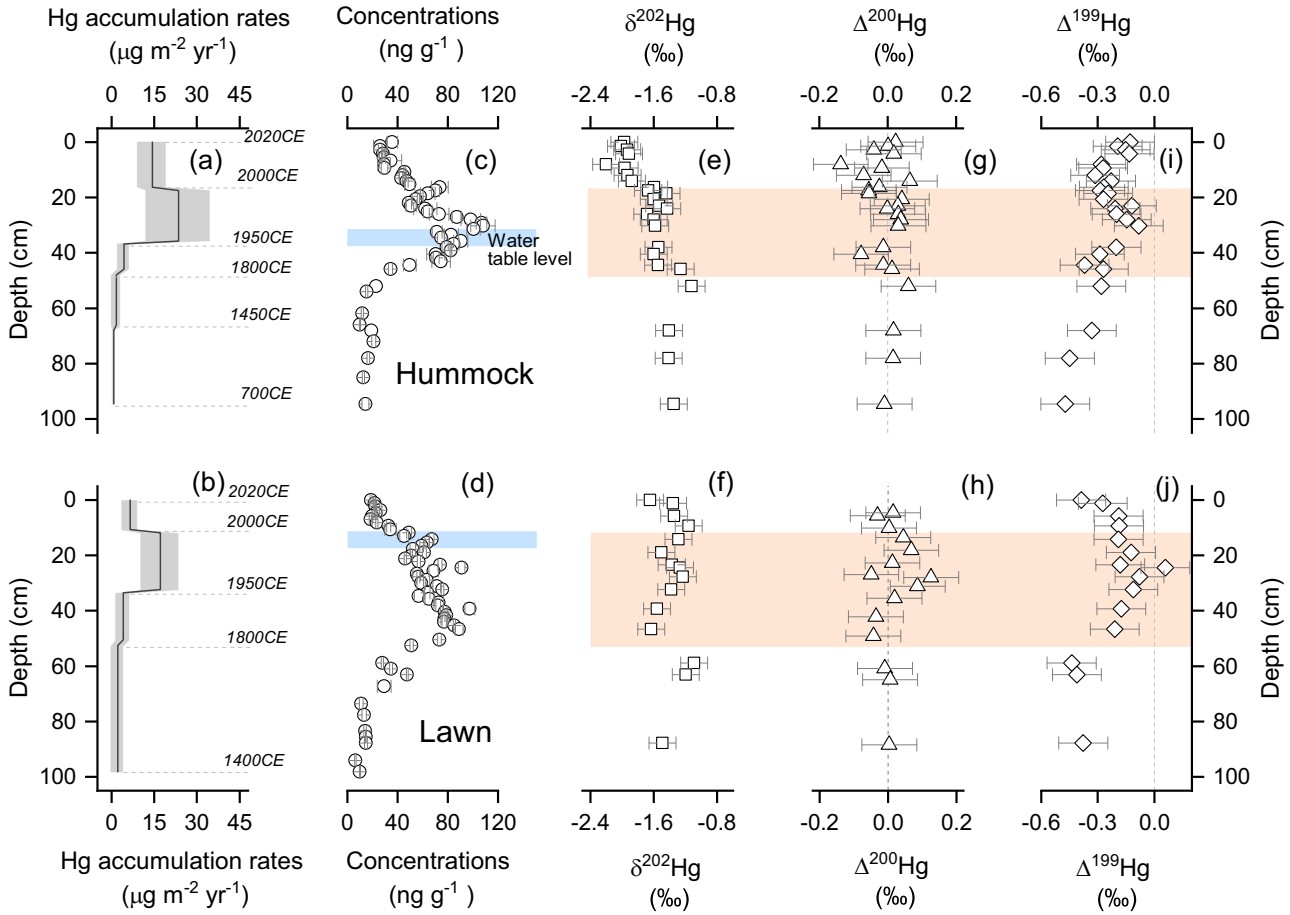

**Fig. 2 | Peat Hg and Hg stable isotope signatures.** Profiles of calculated Hg accumulation rate (**a**, **b** μg m⁻² yr⁻¹), Hg concentration (**c**, **d** ng g⁻¹), δ²⁰²Hg (**e**, **f** ‰), Δ²⁰⁰Hg (**g**, **h** ‰), Δ¹⁹⁹Hg (**i**, **j** ‰) in Degerö hummock peat (upper panel) and lawn peat (lower panel). Grey shaded areas in **a**, **b** represent 1σ of Hg accumulation rate. Blue shaded areas in **c**, **d** represent the average water table level which is 32 ± 5 cm and 12 ± 5 cm during the snow-free period (May to Oct[58]), respectively. Orange shaded areas in **e**–**j** stand for Hg isotope signatures during the period 1800–2000CE. Error bars in **a**–**d** represent 1σ, while those in **e**–**j** represent 2σ uncertainty.

*complex*) are present in the lawn profile (Supplementary Figs. S4 and S5). The Hg concentration in the dominant living hummock species *Sphagnum fuscum* is also significantly higher than in the dominant living lawn species *Sphagnum balticum* (25 ± 0.5 and 18 ± 0.8 ng g⁻¹, respectively, *P* = 0.01, two-tailed T test, Supplementary Table S3). Furthermore, *Sphagnum fuscum* has at least twice the primary productivity (data courtesy from ICOS Sweden, supplementary Fig. S6) and is more decay resistant than *Sphagnum* section *Cuspidata* mosses[39]. This could explain the observation of a higher net peat AR in hummocks than in lawn (0.036 ± 0.012 g cm⁻² yr⁻¹ vs 0.025 ± 0.004 g cm⁻² yr⁻¹, respectively). A higher net peat AR coupled with enhanced net Hg⁰ assimilation in hummock species are likely to be important reasons for greater HgAR in hummock than in lawn at depths corresponding to the period 2000–2020CE (i.e., in the unsaturated zone).

Our findings agree with the suggestion that vegetation types and species composition can modify primary Hg deposition rates[40], although these considerations alone cannot rule out the possibility of a higher Hg loss from the lawn site (please see the following sections for further information on this alternative explanation). Even though the dominant vegetation species and associated Hg deposition rates are different, Degerö hummock and lawn HgAR profiles show similar stepwise increases from the natural background period (i.e., pre-1450CE) to the second half of the 20ᵗʰ century, where peak fluxes are recorded, followed by a decline (Fig. 2a, b). Both HgAR profiles are broadly in line with the trend of rising atmospheric Hg⁰ concentrations

that culminate during the second half of the 20th century in Europe, followed by a sharp drop in emissions and atmospheric concentrations going into the 21st century[41–44].

**Peat Hg stable isotope composition**

Both hummock and lawn profiles are characterized by negative δ²⁰²Hg values of −1.65 ± 0.27‰ (1σ, *n* = 25) and −1.37 ± 0.17‰ (1σ, *n* = 15, Fig. 2e, f), respectively. This is in agreement with preferential uptake of lighter Hg⁰ isotopes by vegetation (supplementary Fig. S8)[19]. The two major Hg sources to peat, i.e., atmospheric Hg⁰ and rainfall Hgᴵᴵ, have distinctly different and conservative Δ²⁰⁰Hg signatures of −0.06 ± 0.02‰ (1σ, *n* = 71,[3,19,45–49]) and 0.16 ± 0.07‰ (1σ, *n* = 55,[3,19,46,49–53]) from Northern Hemisphere (NH) remote areas, respectively. Three samples of atmospheric Hg⁰ at Degerö suggest similar Δ²⁰⁰Hg signatures (−0.10 ± 0.06‰, 1σ, *n* = 3) to atmospheric Hg⁰ values reported in the studies mentioned above. According to the current understanding, MIF of even mass Hg isotopes is relatively conservative over the Earth's surface without being altered during post-deposition transformation processes (e.g., reduction and oxidation[3,53,54]). We assign Δ²⁰⁰Hg values in atmospheric Hg⁰ (−0.06 ± 0.02‰, *n* = 71) and rainfall Hgᴵᴵ (0.16 ± 0.07‰, *n* = 55) based on the composite records from NH remote areas as the end-members for atmospheric deposition at Degerö. The Δ²⁰⁰Hg in both the hummock and lawn peat profiles averages −0.01 ± 0.05‰ (1σ, *n* = 25 for hummock and *n* = 15 for lawn, Fig. 2g, h). Based on atmospheric end-member mixing mass balance calculation for Δ²⁰⁰Hg, plant uptake of Hg⁰ dominates over

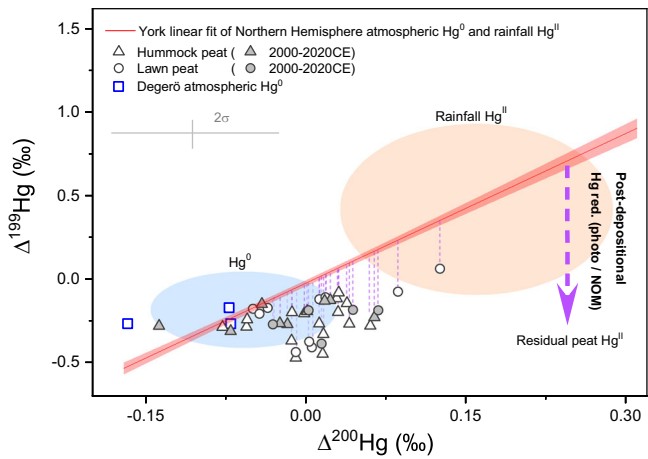

**Fig. 3 | Hg stable isotope signatures in the Northern Hemisphere atmosphere and the Degerö environment.** $\Delta^{199}Hg$ (‰) vs $\Delta^{200}Hg$ (‰) in Northern Hemisphere rainfall $Hg^{II}$ (orange shading, mean + 2σ) and atmospheric $Hg^0$ (blue shading, mean + 2σ), top 1 m hummock peat (black open triangles) and lawn peat (black open circles), and Degerö atmospheric $Hg^0$ (blue open squares). Modern hummock and lawn peat (2000–2020CE) are highlighted by filled symbols. The red line represents York linear fit of Northern Hemisphere atmospheric $Hg^0$ and rainfall $Hg^{II}$[13,19,45–53]. The light red shaded area stands for the 95% confidence band. The downward purple dashed arrow represents the trajectories of residual $Hg^{II}$ after photochemical reduction when bonded to thiols (photo-red.)[20,24], or dark abiotic reduction of $Hg^{II}$ by natural organic matter (NOM-red.)[31]. Neither of the two processes result in significant change in $\Delta^{200}Hg$. The purple dashed lines represent the shift between the York fitting line and peat samples.

precipitation and accounts for a slightly higher proportion of total Hg deposition in the hummock (73 ± 17%, 1σ, n = 25) than in the lawn (66 ± 22%, 1σ, n = 15). These results of dominant plant $Hg^0$ uptake are in agreement with other studies of peatlands[3,37,42], as well as different vegetation ecosystems (e.g., forest and grasslands[4]). $\Delta^{200}Hg$ becomes slightly positive at 1950–2000CE relative to 1800–1950CE in both hummock (increase from −0.03 ± 0.08‰, n = 8, to 0.01 ± 0.08‰, n = 4, 2σ, P = 0.12) and lawn (increase from −0.04 ± 0.08‰, n = 6, to 0.04 ± 0.08‰, n = 2, 2σ, P = 0.06, Fig. 2g, h), which may reflect an enhanced wet deposition during the second half of the 20th century on Degerö (supplementary Fig. S7). This is in line with the increase in precipitation over this part of Sweden since the 1900s, in particular since the mid 20th century[55].

Both hummock and lawn $\Delta^{199}Hg$ profiles shift to more positive values from pre-1800CE to the 2nd half of the 20th century, from −0.47‰ to −0.08‰ and −0.44‰ to 0.06‰ (minimum to maximum value, 2σ = 0.13‰, Fig. 2i, j), respectively. A shift in $\Delta^{199}Hg$ in peat can be explained by either a change in the relative contribution from dry and wet deposition with distinct $\Delta^{199}Hg$ signatures influenced by enhanced anthropogenic emission of Hg to the atmosphere[56], or changes in Hg mobility during post-depositional processes[14]. We do observe a small increase in the assumed conservative $\Delta^{200}Hg$-derived contribution of wet deposition from pre-1800CE to the 2nd half of the 20th century, but this small change in the source contribution alone cannot explain the shift in $\Delta^{199}Hg$. To be more specific, in both hummock and lawn peat, $\Delta^{199}Hg$ values are mostly more negative than the calculated peat $\Delta^{199}Hg$ from the mass balance of the two atmospheric end-members (i.e., NH atmospheric $Hg^0$ and rainfall $Hg^{II}$, see methods, red line in Fig. 3). This provides evidence for post-depositional processes being important contributors to peat $\Delta^{199}Hg$ (e.g., reduction/oxidation of Hg and/or possibly processes associated with the decomposition of peat). A litter decomposition experiment over the course of two-years showed no significant change in the residual $Hg^{II}$ $\Delta^{199}Hg$ (−0.28 ± 0.07‰ to −0.34 ± 0.07‰, 1σ, n = 8)[32], suggesting there would be no significant alteration of $\Delta^{199}Hg$ during decomposition of litter and possibly

organic matter of peat. The observed more negative $\Delta^{199}Hg$ in peat as compared to atmosphere end-members also seems to be in line with the fractionation trajectories of dark abiotic reduction[31], as well as photochemical reduction of $Hg^{II}$[24], both of which leave more negative $\Delta^{199}Hg$ in residual $Hg^{II}$ (Fig. 3). Even though the slope of $\Delta^{199}Hg/\Delta^{201}Hg$ can generally inform about potential reduction processes[17], the associated slopes in hummock and lawn cannot be used to imply the dominant reduction process due to a large uncertainty in determined $\Delta^{201}Hg$ (1.29 ± 0.84, 1σ and 1.75 ± 1.2, 1σ, respectively, supplementary Fig. S9). Some studies have shown evidence of NOM-driven dark reduction being an important process in peat soils[14] and forest soils[32]. If this process is significant in the open boreal peatlands where water table fluctuations create fluctuations in redox potential, one could expect isotopic effects in (i) the associated product $Hg^0$ in the soil gas of the unsaturated zone, i.e., in the peat soil above the peat ground water table, and (ii) in the dissolved gaseous Hg (DGM) of the peat groundwater below the unsaturated zone. In such an assessment, and to further identify Hg post-depositional processes that potentially involved MIF of odd mass, we investigated (i) $Hg^0$ diffusion processes at the atmosphere-peat interface in the unsaturated zone, and (ii) the origin and fate of superficial peat groundwater DGM.

## Hg diffusion at the atmosphere−peatland interface

Over the two year sampling period, the concentrations of $Hg^0$ in the atmosphere average 1.31 ± 0.17 ng m⁻³ (1σ, n = 18, Fig. 4a; Supplementary Table S4), which is at the lower boundary of the concentration range measured at other NH sites mostly spanning the range 1.3 – 1.6 ng m⁻³ [57]. Both hummock and lawn peat soil gas demonstrate similar $Hg^0$ concentrations of 0.43 and 0.48 ng m⁻³ (P > 0.05). These concentrations (0.45 ± 0.12 ng m⁻³, 1σ, n = 38) are consistently below levels observed in the atmosphere. A consistently lower $Hg^0$ concentration in soil gas relative to the atmosphere implies a downward diffusion gradient from the atmosphere into the pore air of the unsaturated zone of the peat soil above the groundwater table. The depth of this unsaturated zone averages 12 ± 5 cm in the lawn (Fig. 2c, d)[58] and is 32 ± 5 cm based on the local elevation of the hummock. The observed gradient from higher $Hg^0$ values in the atmosphere to lower values measured in pore air of unsaturated soils is in line with reports from arctic tundra soil (1.06 ± 0.13 vs 0.54 ± 0.14 ng m⁻³)[34] and mineral forest soils in North America (e.g., 1.16 ± 0.35 vs <0.5 ng m⁻³ below 20 cm in Blodget Forest site)[59].

We use Hg isotope fractionation trajectories between atmospheric $Hg^0$ and peat soil gas $Hg^0$ to deduce the main processes lowering the $Hg^0$ concentration in the peat soil gas[34]. In this study, soil gas $Hg^0$ shows a lower $\delta^{202}Hg$ (−0.09 ± 0.18‰, 1σ, n = 7) and a higher $\Delta^{199}Hg$ (−0.15 ± 0.13‰, 1σ, n = 7) than the atmospheric $Hg^0$ above the peatland surface ($\delta^{202}Hg$ = 0.71 ± 0.19‰, 1σ, n = 3; $\Delta^{199}Hg$ = −0.24 ± 0.06‰, n = 3; Supplementary Table S5). Our determined enrichment factors of $\varepsilon^{202}Hg_{soil\ gas\ -\ atmosphere}$ and $E^{199}Hg_{soil\ gas\ -\ atmosphere}$ are comparable to those in Jiskra et al.[34], which were explained by a depletion of $Hg^0$ in soil gas by dark oxidation governed by NOM. Furthermore, our collected data from Degerö can be fitted by linear regressions (Fig. 4b, c), similar to the experimentally determined isotope trajectories characterized by equilibrium isotope exchange in a closed system, during net dark abiotic oxidation[18]. Both slopes of $\delta^{202}Hg$ vs Hg concentration and $\Delta^{199}Hg$ vs $\delta^{202}Hg$ in our study can be compared with those in Zheng et al.[18] (1.40 ± 0.4 vs 1.04−1.63, and −0.17 ± 0.10 vs −0.11 ± 0.04, respectively). This similarity of slopes further suggests that dark abiotic oxidation of $Hg^0$ to $Hg^{II}$, coupled with equilibrium isotope exchange, is the dominant process regulating $Hg^0$ concentration in the unsaturated zone of the peat, which is a system that appears to be semi-closed in a long-term perspective.

Even if the peat soil gas system in a longer term can be characterized as a semi-closed system, the concentration gradient between atmosphere and soil suggests episodic events of a downward

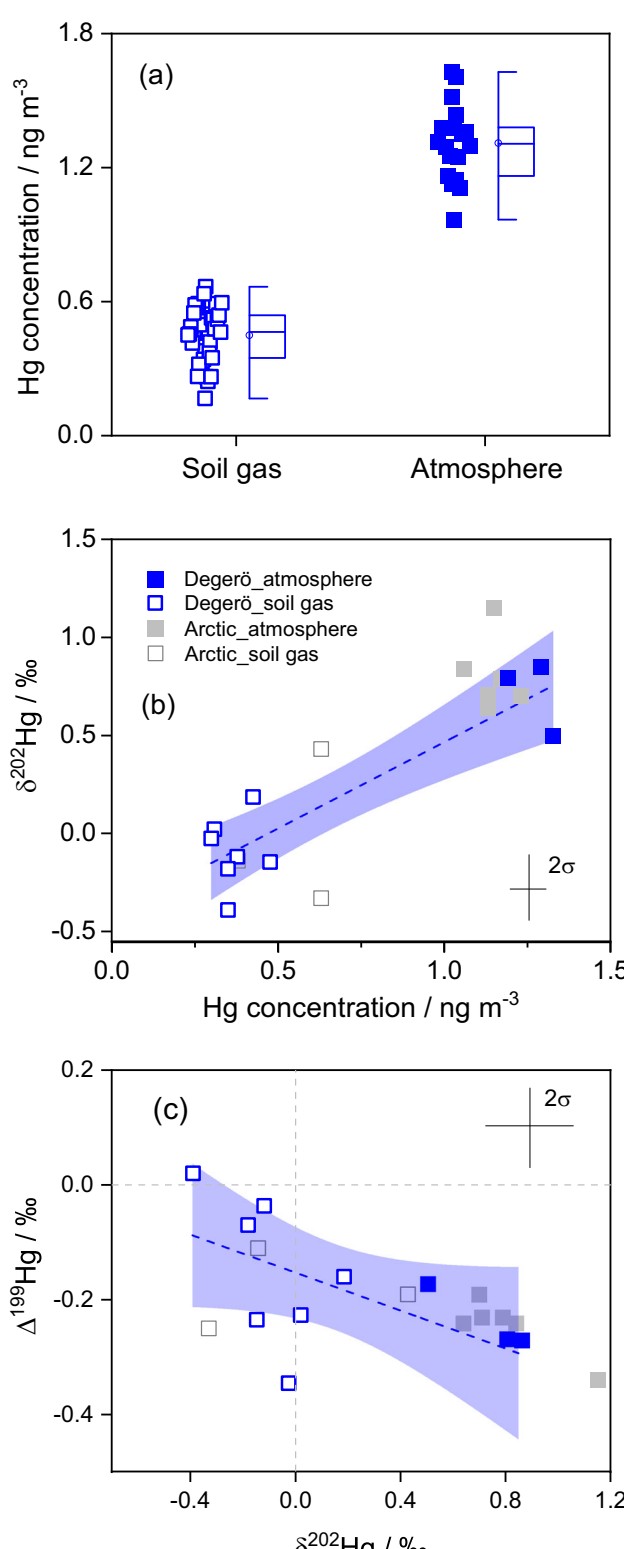

**Fig. 4 | Hg and Hg stable isotope signatures in the atmosphere and soil gas.** $Hg^0$ concentration gradient between atmosphere and peat soil gas integrated from both hummock and lawn sites in 2020 and 2021 **a**. $\delta^{202}Hg$ vs Hg concentration **b** and $\Delta^{199}Hg$ vs $\delta^{202}Hg$ **c** in Degerö atmosphere (blue filled squares) and peat soil gas samples from the unsaturated zone above the groundwater table level (blue open squares). Blue dashed lines represent the regression of York linear model **b**, **c**. Grey filled and open squares represent atmosphere and peat soil gas samples from Arctic tundra, respectively[34]. The light blue shaded areas denote the 95% confidence intervals. Error bars represent the analytical precision (2σ) based on multiple analyses of procedural standards.

net flux of $Hg^0$ to the gas phase of the unsaturated zone of the peat. By use of Fick's first law, we calculated the potential vertical diffusion flux of $Hg^0$ from the lower atmosphere into the unsaturated peat layer (Fs, ng m⁻³ d⁻¹, supplementary Text S2). We estimate a $Hg^0$ downward diffusive net flux of 0.0001 ng m⁻² d⁻¹. This calculation does not include data on potential convective flow or plant mediated transport of Hg at the soil surface. Our estimated flux is much lower than the downward diffusion of 0.2 ng Hg m⁻² d⁻¹ reported from atmosphere to Northern American forest soil[59]. Even if the estimated annual downward flux of $4 * 10^{-5}$ μg m⁻² yr⁻¹, indirectly caused by dark abiotic oxidation, is negligible as compared to the total dry deposition of Hg to the peatland, the net downward flux of $Hg^0$ clearly rules out the possibility of any net evasion of Hg from the peat soil through the air-filled pores of the unsaturated zone during the study period. The downward flux of Hg also lends support for excluding dark reduction of Hg by NOM in the peat soil as a significant process behind the $\Delta^{199}Hg$ anomaly (Fig. 3). The low diffusive flux of $Hg^0$ further supports the characterization of the peat unsaturated zone as a semi-closed system.

### Origin and fate of dissolved gaseous mercury (DGM)

During summer the DGM concentrations average 77 pg L⁻¹ (corresponding to 77 ng m⁻³ of the aqueous phase) in the Degerö peat groundwater 0 – 15 cm below the groundwater surface (Fig. 5a). This suggests supersaturated conditions relative to the average concentration of $Hg^0$ in the air-filled pores (gas phase) of the unsaturated zone of the peat ($0.45 \pm 0.12$ ng m⁻³), as well as to the atmospheric $Hg^0$ concentration of $1.31 \pm 0.17$ ng m⁻³ just above the peatland. The $Hg^0$ saturation level in the groundwater is 1500% and 4300% relative to the atmosphere and the peat soil gas phase, respectively (supplementary Text S3). The DGM concentrations reach a maximum just below the groundwater table and then decline with depth to 55 cm below the water table (Fig. 5a). Even though the shallow groundwater DGM is oversaturated in relation to $Hg^0$ in the atmosphere, the fact that the soil gas $Hg^0$ concentration is lower than that in the atmosphere (discussed above) suggests that there is no net diffusion of DGM to the atmosphere via the air-filled soil pores in the peat. However, at this stage we cannot exclude a potential upward diffusion of DGM along the water-saturated pores that exist side-by-side with air-filled pores in the unsaturated zone of the peat, or through the aerenchymatous tissues of vascular plants. Some of this Hg would diffuse into soil gas to be oxidized by NOM in the unsaturated zone similar to the fate of peat gas $Hg^0$ originating from the atmosphere. The DGM in the groundwater also diffuses downwards along the established concentration gradient. Even if the increase in reduced organic sulfur species below the annual water table (Supplementary Figs. S10, S11) reveals a more reducing environment in the water saturated zone, the DGM profile could reflect a slow movement (diffusion) of DGM downwards if the rate of $Hg^0$ oxidation exceeds the rate of $Hg^{II}$ dark reduction. It has been shown that reduced NOM possess a much higher potential to both reduce $Hg^{II}$ and back-oxidize $Hg^0$ than oxidized NOM[60]. The dark reduction of $Hg^{II}$ is expected to be very slow due to its exceptionally strong complex formation with thiol groups in NOM[61], and possible formation of the only slightly soluble mineral β-HgS (metacinnabar) in the sulfidic environment of peatland soils in the region[62].

Another constraint on the origin and fate of DGM in peat groundwater is provided by the composition of Hg isotopes. A significant difference in conservative $\Delta^{200}Hg$ between DGM measured below the ground water table and peat soil gas Hg measured above the same groundwater table ($P < 0.05$, Fig. 5b, Supplementary Table S6) indicates that upward diffusion of DGM to the unsaturated zone in the peat is unlikely. In addition, no MIF has been observed during volatilization of $Hg^0$ from solution into the gas phase[63], while $E^{199}Hg_{DGM\text{-} soil\ gas}$ in our study is $-0.33 \pm 0.12$‰ (1σ). Furthermore, we do not find any evidence of DGM in the peatland groundwater being

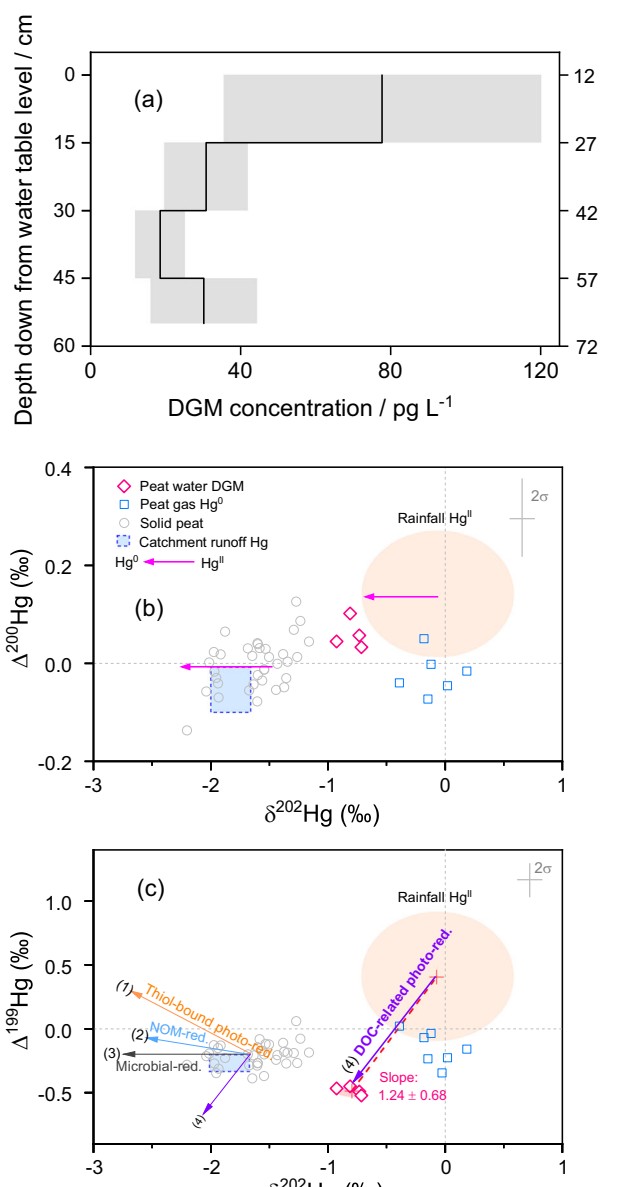

**Fig. 5 | Dissolved gaseous mercury (DGM) concentration, isotope signatures and potential origins. a** Depth profile of DGM concentration. The y-axis is relative to the groundwater table level and the secondary y-axis is relative to the lawn peat surface. **b** $\Delta^{200}$Hg vs $\delta^{202}$Hg and **c** $\Delta^{199}$Hg vs $\delta^{202}$Hg in Northern Hemisphere rainfall Hg$^{II}$ (orange shading, mean + 2σ), peat gas Hg$^0$ from both lawn and hummock (blue open squares), solid peat from both lawn and hummock (top 50 cm corresponding to the depth of DGM sampling below the groundwater table for Hg isotope analysis, grey open circles) and DGM from lawn site (pink open diamonds). Blue dash box represents North American peatland runoff Hg isotope data from both high streamflow and low streamflow (mean + 1σ, $n = 24$) in Woerndle et al. [37], which has similar ranges of mass dependent fractionation and odd-mass independent fractionation in peat soils to our study. Pink horizontal arrows in **b** represent trajectories of $\Delta^{200}$Hg vs $\delta^{202}$Hg for both biotic and abiotic reduction given that even-mass independent fractionation is insignificant [31]. Arrows in **c** represent trajectories of photoreduction of Hg$^{II}$ complexed by thiols in NOM (orange, (1) [20]), dark reduction by NOM (blue, (2) [31]), microbial reduction (grey, (3) [30]), and photo-reduction in the presence of dissolved organic carbon (DOC, purple, (4) [26]), respectively. The pink dashed line shows the slope of 1.24 ± 0.68 in $\Delta^{199}$Hg vs $\delta^{202}$Hg between rainfall Hg$^{II}$ and DGM.

produced from Hg$^{II}$ stored in the peat, giving rise to a current source of Hg$^0$ diffusing through the peat unsaturated zone and back to the atmosphere. However, it cannot be ruled out that DGM evasion to the atmosphere may be of importance during submerged conditions

when the water table rises to the peat surface e.g., late fall, winter, and early spring. Such DGM fluxes to the atmosphere have been reported from freshwater lakes [64,65].

Peat groundwater DGM $\Delta^{200}$Hg even-MIF lies between rainfall and peat soil data (Fig. 5b), pointing to a possibly mixed contribution from Hg$^0$ produced by reduction of Hg$^{II}$ stored in the solid peat, peat groundwater and of Hg$^{II}$ in rainfall. The $\Delta^{200}$Hg-based mass balance derived from Monte Carlo simulations shows that 48% of this DGM originates from peat groundwater Hg$^{II}$ (27–64%, IQR), and 52% from rainwater Hg$^{II}$ (35–73%, IQR). The high uncertainties in the DGM contribution from these two sources warrant an investigation of other Hg isotope signatures. The product of reduction (i.e., Hg$^0$) is generally enriched in lighter isotopes (i.e., more negative $\delta^{202}$Hg [17]). At Degerö the DGM $\delta^{202}$Hg is more positive than $\delta^{202}$Hg in solid peat and peat groundwater (Fig. 5b), excluding peat soil as a dominant source of DGM in peat groundwater. In contrast, the trajectory of $\delta^{202}$Hg between rainfall and DGM is in line with the trajectories of abiotic and biotic reduction (Fig. 5b), indicating that Hg$^{II}$ in rainfall may be the major source of DGM. Notably the concentration of Hg$^0$ in local rainfall (27 ± 5.7 pg L$^{-1}$, 1σ, $n = 6$, Supplementary Table S7) is half of that in the surface peat groundwater DGM collected a few hours after the rain event (54 ± 1.8 pg L$^{-1}$, $n = 2$). Given equal water volumes of rainwater and groundwater, Hg in rainwater could at the most account for half of the quantity of Hg$^{II}$ collected in the superficial peat groundwater on the same day. Even though no rainfall DGM isotope signatures have ever been reported, lower $\delta^{202}$Hg and lower $\Delta^{199}$Hg relative to those in rainfall Hg$^{II}$ (Fig. 5c) could be inferred from the rainfall Hg$^{II}$ and particle-bound Hg isotope compositions under photochemical reduction [53]. While likely more than one-half of the groundwater DGM can be explained by rainfall DGM, the rest might be attributed to reduction of rainwater Hg$^{II}$ after deposition to peat. Our observation of low $\delta^{202}$Hg in the DGM compared to rainwater (Fig. 5b), is in line with microbial and abiotic reduction of rainwater Hg$^{II}$, which produces Hg$^0$ with lighter isotopes [30]. More negative $\Delta^{199}$Hg in DGM than rainwater, solid peat, and peatland runoff excludes the possibility of significant contribution from dark reduction of NOM-bound Hg$^{II}$, which leads to a more positive $\Delta^{199}$Hg in the product [31].

The residence time of fresh rainwater in the first 1–3 cm below the peatland surface (in principal the length of living mosses indicated by the green color) can last for hours to days during periods of precipitation and downward water flux [66–68], potentially enabling photoreduction of Hg$^{II}$ dissolved in rainwater droplets physically attached to and/or encapsulated in cavities of living moss structures. The trajectory line of $\delta^{202}$Hg vs $\Delta^{199}$Hg between rainfall Hg and DGM reveals a slope of 1.24 ± 0.68, which is in an agreement with the experimental trajectory describing photoreduction of Hg$^{II}$ dissolved in the aqueous phase in the presence of DOC (dissolved organic carbon, 1.15 ± 0.07 [26], Fig. 5c). Thus, the DGM isotopic signature suggests that Hg$^{II}$ provided by rainfall and exposed to sunlight is likely an important source of DGM in peat groundwater. Light-driven Hg$^{II}$ reduction in systems with DOC is several orders of magnitude faster than dark reduction in the presence of DOC [31,69]. Our suggestion of photoreduction of rainwater Hg$^{II}$ is in line with isotopic data from agricultural soils [70], in which O/N functional groups were suggested to be involved in the initial complexation of Hg$^{II}$ after deposition. As a consequence of the weaker Hg-O/N bond, rates of Hg$^{II}$ reduction are greater than when Hg$^{II}$ forms complexes with the chemically much stronger bonding thiol functional groups [61,69]. Bonding of Hg$^{II}$ to thiols is expected to occur within minutes to hours and completely dictate dark reduction rates of Hg$^{II}$ below the immediate surface of the peatland soils [7]. Our results are consistent with the hypothesis that the rainfall pool of Hg$^{II}$ in the peatland could be more susceptible than peat Hg$^{II}$ to photoreduction [37]. Overall, we judge that it is likely to be rainfall Hg$^{II}$, instead of Hg$^{II}$ from decomposing peat, that is the dominant source of DGM in the peat groundwater.

## Rates of Hg loss through photoreduction at the peat surface since 1800CE

The dominance of Hg⁰ oxidation in the unsaturated zone of peat soil and a non-significant production of DGM from Hgᴵᴵ stored in the solid peatland ending up in the peat groundwater suggests that NOM-driven dark reduction is unlikely to be a significant process contributing to the negative peat $\Delta^{199}$Hg observed in peat soil (Fig. 3). We, therefore, consider the potential of photon-driven Hg⁰ formation and loss to explain this isotopic shift. Increases in Hg photoreduction of Hgᴵᴵ at lake water surfaces[56,71] and/or in recently fallen rain[56] have been used to explain $\Delta^{199}$Hg anomalies in lake sediments. Photochemical reactions active at the surface of living vegetation of the open Degerö peatland during the snow-free period (May to Oct)[58], can lead to reduction of Hgᴵᴵ, which is generally characterized by a fractionation trajectory with negative $\Delta^{199}$Hg in residual vegetation Hgᴵᴵ when complexed by thiol functional groups[24]. Our observed lower peat $\Delta^{199}$Hg (Fig. 3) is in good agreement with the fractionation trajectory of photochemical reduction of Hg on the surface living vegetation, potentially leading to Hg loss.

We calculated the Hg⁰ loss and emission from the peat surface to the atmosphere based on Rayleigh fractionation model[72] and the isotopic enrichment factors for photochemical reduction of Hgᴵᴵ on foliage ($E^{199}$Hg$_{reactant/product}$ = 0.49[24], supplementary Text S4). The proportion of the calculated photoreductive Hg loss since 1800CE is similar in hummock and lawn, with $28 \pm 16\%$ ($4.5 \pm 3.8 \,\mu g\, m^{-2}\, yr^{-1}$) and $27 \pm 20\%$ ($3.5 \pm 4.2 \,\mu g\, m^{-2}\, yr^{-1}$), respectively. The size of this proportional loss is also in line with the estimates on Hg⁰ re-emitted from foliage by photoreduction[24].

An additional peat Hg loss pathway is through discharge export of Hgᴵᴵ complexed by dissolved organic matter from the Degerö catchment area. This loss was estimated to be $1.6 \pm 0.2 \,\mu g\, m^{-2}\, yr^{-1}$ for the period 2009–2014CE[12]. The Hg export via streamflow mainly originates from the peatland system with a minor contribution from the upland mineral soils (covering 30% of total catchment area)[12]. Thus, rainfall, snowmelt and peat groundwater are the major contributors of Hg in streamflow[12,37]. The ¹⁴C dating of the organic matter in streamflow from the Degerö mire shows that this organic matter is less than half a century old[73], suggesting that it is Hg deposited during the last 50 years that is possibly mobilized by water flow through the peatland system. This estimated Hg loss in streamflow corresponds to less than half of the losses generated from photoreduction, which is the most important process by which Hg is lost from the open peatland.

The absolute amount of photoreductive Hg loss between 2000 and 2020CE was similar in hummock and lawn ($2.9 \pm 2.1$ vs $2.2 \pm 0.7 \,\mu g\, m^{-2}\, yr^{-1}$, Supplementary Fig. S12). The proportional and absolute photoreductive Hg loss in lawn during peak HgAR periods (1950–2000CE) was not statistically different from hummock ($25\% \pm 22\%$, $n = 6$ vs $31\% \pm 13\%$, $n = 8$, $P = 0.5$, and $5.2 \pm 5.6$ vs $8.0 \pm 3.5 \,\mu g\, m^{-2}\, yr^{-1}$, $n = 6$ vs 8, $P = 0.6$, Supplementary Fig. S12). This indicates that the lower HgAR since 1950CE in lawn, as compared to hummock ($17.1 \pm 10.8$ vs $24.6 \pm 14.5 \,\mu g\, m^{-2}\, yr^{-1}$) is not due to a higher Hg loss, but rather to less Hg deposition. This emphasizes the importance of vegetation composition and associated primary productivity in the transfer of atmospheric Hg to terrestrial environments[40]. Our study suggests that photoreductive Hg loss dominates Hg mobility in peatlands, accounting for approximately 30% of the annual Hg deposition (Fig. 6).

## Environmental implications

Mercury is deposited onto open peatlands by two predominant processes: (i) direct plant uptake of atmospheric Hg⁰, and (ii) input of atmospheric Hgᴵᴵ by wet deposition (e.g., rainfall). The plant uptake of atmospheric Hg⁰ accounts for the major input, approximately 70%, to the open boreal peatland at Degerö. About 30% of the total Hg input is released back to the atmosphere as Hg⁰, produced by the

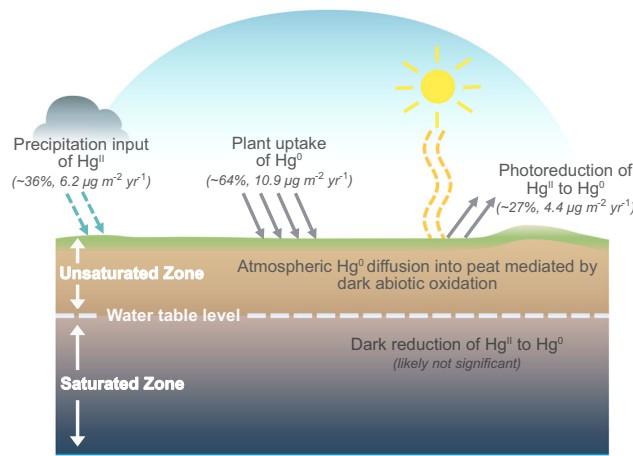

**Fig. 6 | The new conceptual model of Hg deposition and mobility in a boreal peatland based on the findings of this study.** Precipitation input of Hgᴵᴵ and plant uptake of Hg⁰ account for ~36% and ~64% of total Hg deposition (6.2 vs $10.9 \,\mu g\, m^{-2}\, yr^{-1}$) since 1800CE, respectively. Atmospheric Hg⁰ diffuses into the unsaturated zone of the peat, as driven by the process of dark abiotic oxidation, even though this flux is negligible as compared to the input of Hg in rainfall and plant uptake. Photoreduction of Hgᴵᴵ to Hg⁰ and subsequent evasion at the peat surface corresponds to 27% of the total Hg deposition ($4.4 \,\mu g\, m^{-2}\, yr^{-1}$). In contrast, the dark reduction of Hgᴵᴵ to Hg⁰ in the saturated zone of the peat is likely not significant compared to the photoreduction process.

photoreduction of Hgᴵᴵ at surfaces of the peatland vegetation. The dominance of net plant uptake of atmospheric Hg⁰ over net rainfall Hgᴵᴵ deposition to the vegetated surface of peatlands might be partially due to Hgᴵᴵ in precipitation being more readily available for photoreduction and subsequent evasion back to the atmosphere than the Hgᴵᴵ associated with plant tissues. The relative importance of Hgᴵᴵ photoreduction in rainfall can be explained by the abundance of organic R-O/N functional groups at peat and vegetation surfaces providing a weaker bond to Hgᴵᴵ to compete with the reduction process, during the time window required for Hgᴵᴵ to re-arrange to stronger bonding RS functional groups[31,61,70]. Once bound to the stronger RS functional groups, Hgᴵᴵ is less susceptible to reduction. Our results on detailed post-depositional Hg redox processes provide new information that helps to explain the poorly understood mechanisms behind Hgᴵᴵ reduction and Hg⁰ re-emission[5], particularly in open peatland ecosystems (i.e., no tree cover)[24].

Mercury concentrations and isotope signatures highlight the oxidation capacity of organic matter in the air-filled pores of the peat above the groundwater table. This lowers the Hg⁰ concentrations in the soil gas relative to the atmosphere. While the downward diffusion flux appears to be negligible as compared to total Hg deposition to the peatland, it clearly rules out upward net diffusion of Hg⁰ from the peatland air-filled sub-layers to the atmosphere. Below the groundwater table of the peat, the concentration of DGM is supersaturated in relation to both the gas phase of the unsaturated zone of the peat, as well as the atmosphere. The main source of the DGM is likely to be rainwater Hgᴵᴵ photoreduction before and/or after deposition to the peatland surface, rather than dark reduction Hgᴵᴵ in the peat soil. This means we have not found any reduction mechanisms that would significantly redistribute Hg in gaseous form once it is incorporated into the peat profile and thereby change the peat Hg archive during decomposition processes. There is a small amount of the previously deposited Hg (10%) that is exported downstream together with dissolved organic matter in stream runoff. Peat soil Hg isotopes are not likely to be significantly altered by this stream runoff based on the similarity of Hg

isotope composition in boreal forest runoff and soils[74]. The processes of Hg transformation we unravel here would constrain the mobility of Hg while the peat OM slowly decays below the groundwater table, and thus the possibilities for mobilization of Hg deposited from the atmosphere in earlier decades or centuries. Compared to tree-covered forested ecosystems[14], dark reduction of $Hg^{II}$ by NOM appear less important in these types of open (tree-less) peatlands where there is more sunlight to promote photoreduction of $Hg^{II}$ at the surface. The dominant peat Hg loss caused by photoreduction can likely reduce Hg methylation rates by decreasing the size of this weakly-bond $Hg^{II}$ pool which is expected to have a high availability for methylation. Our study highlights that it is the peat surface, instead of peat sub-layers, where main Hg loss occurs. We do, however, suggest that the redox transformation processes during peat decomposition are not significant enough to disqualify peat soil to be a reasonable archive of long-term atmospheric Hg deposition patterns.

## Methods

### Study site

This study was conducted at the Degerö Stormyr, a Sphagnum-dominated minerotrophic open peatland, on areas without tree cover (supplementary Fig. S1-a). The peatland covers two-thirds of the 6.5 km² catchment area which is 270 m above sea level in the Kulbäcksliden Research Park (64°11′N and 19°33′E). This is located in the municipality of Vindeln municipality, Västerbotten province, Sweden. The climate at Degerö peatland is cold temperate humid, with mean annual values of 523 mm for precipitation and +1.2 °C for temperature (data from 1961 to 1990[75]). Mean temperatures in July and January are +14.7 °C and −12.4 °C, respectively. The vegetation growing season is from May to October (156 ± 15 days based on 2001–2005[58]). The rest of the year is characterized by snow cover, which in general reaches a depth of 0.6 m. Previous work on Hg conducted in this peatland includes the influence of sulphate concentration on peat pore water Hg methylation[76–78], and Hg flux measurements[12,79,80]. The Degerö peatland surface is dominated by lawns with minor occurrence of hummocks.

### Solid peat coring and sub-sampling

One 3 m-long peat sequence (DEG20-PH01A) was collected from the Degerö hummock site in July 2020 (supplementary Fig. S1-b). Five meters away from DEG20-PH01A, another 350 cm-long peat sequence (DEG20-PL01A) was sampled in the lawn site. A PVC tube of 15 cm internal diameter and 50 cm length was used for the top 50 cm peat collection[12]. For the deeper layers, a stainless steel Russian corer with 7.5 cm internal diameter and 100 cm length was used[81]. Three other hummock peat sequences (ca. 300 cm) and two other lawn peat sequences (ca. 350 cm) at the same sites were collected and stored as archive samples. Peat cores were described for some basic information (e.g., length and color) and then wrapped in plastic film before placement in PVC tubes for transport to the Swedish University of Agricultural Sciences (SLU, Umea campus, Sweden). Cores were frozen and subsequently sliced at roughly 1 cm resolution for the top 50 cm of peat and then at 2 cm resolution for the rest of the core. Each new slice was cleaned with MilliQ water, edges removed and subsampled for further analysis following well-established protocols[82,83]. The dimension of the largest subsample of each slice was measured using a Vernier caliper to obtain the volume for calculating the dry bulk density and to estimate the cut loss between each slice. Subsequently, the largest sub samples were dried for geochemical analysis using the freeze-dryers at SLU (Christ Alpha 1-4 LSC Plus; ScanVac CoolSafe). In this paper, we focus on the top 1 m to understand the Hg geochemical cycle in the zone where the water table fluctuates and the acrotelm (above peat groundwater table level) transitions into the catotelm (below groundwater table level).

### Radiocarbon dating and age models

In total 27 plant macrofossil samples from hummock profiles and 32 from lawn profiles, were selected for radiocarbon analyses following established protocols[84,85]. All the selected samples were prepared and analyzed for ¹⁴C at the Ångström laboratory of Uppsala University (Uppsala, Sweden). Fourteen hummock and thirteen lawn samples were dated to a post-bomb period, whose ages were calibrated using the NH Zone 1 calibration curve provided by Calibomb software of Queen's University, Belfast[86–88]. The age models for hummock peat profiles (27 dates) and lawn peat profiles (32 dates) were generated from post-bomb calibrated ages and pre-bomb ¹⁴C results using the Bacon model (calibration curve IntCal20) with the 'rbacon' package in R software (https://CRAN.R-project.org/package=rbacon)[89]. Details on the dated material, radiocarbon ages and calibrated ages are shown in Supplementary Table S1.

### Plant macrofossil analysis

A total of 25 and 26 fresh samples from the top 1 m hummock and lawn profiles, respectively, were chosen for macrofossil identification. Macrofossil samples were warmed in 10% NaOH and sieved (mesh diameter 180 µm). Macrofossils were identified using a binocular microscope (×10 – ×50) based upon modern type material. Identifications were also made with reference to Michaelis, (2011)[90] for *Sphagnum* mosses. Volume abundances of all components are expressed as percentages with the exception of *Andromeda polifolia* seeds, *Eriophorum vaginatum* spindles, *Carex* spp. nutlets, *Sphagnum* spore capsules, *Cristatella mucedo* statoblasts and macrofossil charcoal fragments, which are presented as the number (n) found in each of the subsamples. Zonation of the macrofossil diagram was made using psimpoll 4.27[91], using the optimal splitting by information content option.

### XANES analysis

To obtain the information on sulfur species for examining the reduction/oxidation condition in peatland, Sulfur K-edge X-ray absorption near edge structure spectra (XANES) were collected and analyzed at Beamline 4B7A in Beijing Synchrotron Radiation Facilities (BSRF)[7,92]. Briefly, spectra were obtained from 12 freeze-dried samples from hummock profile in fluorescence mode at ambient temperature under high vacuum ($10^{-8} – 10^{-6}$ mbar). The storage ring was operated at 2.5 GeV with a ring current of 250 mA. A fixed double-crystal monochromator with Si(111) crystals was used to monochromatize the white beam. Scans were taken at the energy range of 2462–2500 eV with a step size of 0.2 eV. Data averaging, normalization, and Gaussian curve deconvolution were conducted using Athena, WinXAS, and Microsoft Excel (Supplementary Figs. S10 and S11).

### Peat soil gas, atmosphere, peat water, and rainfall sampling

We sampled peat soil gas and atmosphere in both Degerö hummock and lawn for the measurements of Hg concentration and Hg stable isotopes over two summers from 2020 to 2021 (Supplementary Fig. S1-c; Fig. S13). Peat soil gas and atmosphere were continuously sampled by a pump using PFA tubing (1/4" outer diameter (OD), 5/32" inner diameter (ID), Savillex) with a filter (Teflon) mounted at the gas inlet of each tube to prevent the entry of moisture. Gas inlets were placed at depths of −15 cm, −10 cm, and +25 cm relative to the living peat surface for hummock, lawn, and atmosphere sites, respectively. To lower the sampling flow rate and not cause potential isotopic fractionation, each gas inlet for peat soil gas Hg isotope analysis was further subdivided into three tubes, each with sampling rates of 0.15 LPM. Iodated activated carbon traps were used to collect peat soil gas over the period of five to seven weeks for Hg isotope analysis, while gold traps (Teflon) were used for concentration analysis with a sampling duration of hours. Gold traps were further used to test peat soil gas $Hg^0$ concentration at sampling rates from 0.05 to 0.42 LPM. We did not find a significant difference between these different rates (0.48 ± 0.05 ng m⁻³,

$n = 10$, $R^2 = 0.11$). This indicates that the sampling rates were sufficiently low to avoid drawing in air from above the peat.

To collect peat groundwater for DGM isotope analysis, three perforated PVC tubes with plugs at each end (10 cm ID), were buried below the lawn peat surface to serve as groundwater reservoirs. Each reservoir tube was 3 m long and buried with one end at −30 cm below the lawn peat surface, and the other end at -50 cm. At both ends of each PVC reservoir tube, there was a vertical, 1 m-long PVC access tube with 2 cm ID. This extended above the peat surface allowing peat water to be easily pumped from the lower end of each of the 3 m-long reservoir tubes (Supplementary Fig. S1-d). The tubes were buried in the peat one-month prior to the start of sampling. To sample the peat groundwater, the 1 m-long vertical access tube at the deeper end of the reservoir was connected to a peristaltic pump. Peat water was pumped into a 20 L glass bottle (Sarl Ellipse, France) wrapped in black plastic to block sunlight. When fully filled, the bottles were immediately transported to the Östvallen laboratory, which is a 20 min drive from the field site. We collected three to six full bottles of peat water on a daily basis from 23rd June to 13th July 2021.

Peat groundwater for DGM concentration analysis was sampled using a hollow Teflon probe that was connected to a 250 ml Teflon PFA vessel and a rotary vane pump[12]. This peat groundwater was collected at four sites in the Degerö peatland at depths below the water table of 0 – 10, 15 – 25, 30 – 40, and 45 – 55 cm. The concentration measurements were made during the same sampling period as for the DGM isotope analysis—namely between 23rd Jun and 13th Jul 2021. All peat groundwater samples for Hg concentration and isotope analysis were well protected from sunlight during sampling, transport, and extraction to eliminate photolytic reactions.

Rainfall samples for DGM concentration analysis were collected in an open area of Östvallen laboratory away from any possible contamination sources (e.g., engines/cars). The collection system consists of an inclined acid-washed Teflon-coated black plate connecting to a clean funnel at the lower edge of the plate. A 500 ml glass bottle was placed at the outlet of the funnel. Three rainfall samples were collected within 1 h during rain events on July 5th and 14th 2022 (supplementary Table S7). As a comparison, two peat surface groundwater samples (0 – 10 cm) were also sampled for DGM concentration analysis at Degerö on 14th July 2022.

## Pre-concentration of dissolved $Hg^0$ for Hg stable isotope analysis

Due to the low $Hg^0$ concentration in peat groundwater, approximately 2000 L of peat water was collected for pre-concentration from 23rd June to 13th July 2021, enabling four aliquots of DGM isotope measurements (i.e., 10 ng $Hg^0$ per measurement). We adapted a rainfall Hg purging method described in Jiskra et al.[54] to our peat groundwater DGM extraction system (Supplementary Fig. S14). We started to pre-concentrate 16 L peat groundwater with 4 L headspace within 1 h of sampling at the Östvallen laboratory using 20 L glass bottles (Sarl Ellipse, France) over a period of 3 h. The GL45 two-port PFA Teflon cap (Savillex) was used to replace the GL45 PFA Teflon cap (Savillex) and guided a 55 cm long, 6 mm outer diameter, 3 – 4 mm inner diameter Pyrex bubbling post with a 1 cm-long P3 porosity frit (Saveen Werner, Sweden). The second port on the GL45 cap hosted a 1 m long, 6 mm OD FEP tube that was connected to a Teflon filter, then a soda lime filter, followed by a carbon trap filled with 400 mg of iodated activated carbon (Brooks Rand) that collected peat DGM for Hg isotope analysis. A flow meter (1 L/min, Masterflex™ 65 mm) was installed after the carbon trap and connected to a pump. All the glassware was cleaned with reversed aqua regia at 100 °C. The Teflon components were cleaned by 2% $HNO_3$. Prior to pre-concentration, we tested 0 ng Hg in the blank of the sampling lines without peat water (Supplementary Fig. S14).

## Hg concentration measurements and Hg accumulation rate calculation

Once rainfall and peat groundwater samples were collected, the DGM was immediately analyzed for concentration on a Tekran 2537X (Supplementary Fig. S15). This was calibrated at least once a week. Prior to analysis, we tested 0 ng Hg in the blank of the sampling line.

Freeze-dried peat samples were analyzed for total Hg (THg) concentration on a combustion cold vapor atomic absorption spectrometer (CV-AAS, Milestone DMA-80) at the Swedish University of Agricultural Science, Uppsala, Sweden. The analytical performance of the DMA-80 was assessed by multiple measurements on reference materials, NIST 1515 (Apple leaves) and BCR 482 (Lichen). Results were not statistically different from the certified values, with Hg concentrations of $42.9 \pm 3.9$ ng g$^{-1}$ (1σ, $n = 129$, certified $43.2 \pm 2.3$ ng g$^{-1}$) for NIST 1515, and $458 \pm 13$ (1σ, $n = 120$, certified $480 \pm 20$ ng g$^{-1}$) for BCR 482.

Hg accumulation rate (HgAR, μg m$^{-2}$ yr$^{-1}$, Eq. 1) in sample $i$ was obtained by Hg concentration (ng g$^{-1}$), density (g cm$^{-3}$), thickness (cm) and age interval (yr).

$$HgAR_i = Hg\ concentration_i \times density_i \times \frac{thickness_i}{age\ interval_i} \quad (1)$$

## Hg isotope measurements

Samples of solid peat and carbon traps were processed using a combustion method adapted from Enrico et al.[93]. Hg released from the combustion procedure was collected with 40% inverse *aqua regia* solutions. Following extraction, the Hg stable isotope compositions of 40 solid peat samples from the top 1 m section (25 hummock and 15 lawn), three atmosphere, seven peat soil gas and four peat water DGM samples were determined from 20% (v/v, diluted from 40%) inverse *aqua regia* solutions using cold-vapor multi-collector inductively coupled mass spectrometry (CV-MC-ICP-MS, Nu, ETHZ). Sample isotopic ratios were corrected for mass bias by sample-standard bracketing using NIST 3133[94]. Results are reported as δ-values in per mil (‰) representing Hg mass dependent fractionation by reference to NIST 3133 (Eq. 2).

$$\delta^{XXX}Hg = \left\{ \frac{(^{XXX}Hg/^{198}Hg)_{sample}}{(^{XXX}Hg/^{198}Hg)_{NIST3133}} - 1 \right\} \times 1000 \quad (2)$$

MIF is calculated based on the deviations of δ-values from the theoretical MDF (Eq. 3).

$$\Delta^{XXX}Hg = \delta^{XXX}Hg - \beta \times \delta^{202}Hg \quad (3)$$

where XXX stands for 199, 200, 201 and 204. Symbol $\beta$ is 0.2520, 0.5024, 0.7520, and 1.493 for $^{199}$Hg, $^{200}$Hg, $^{201}$Hg, and $^{204}$Hg, respectively.

The quality control of Hg isotope measurements is assessed by analyzing ETH-Fluka and procedural standards (Apple leaves, NIST 1515, $n = 5$, Supplementary Table S8). ETH-Fluka displayed $\delta^{202}$Hg and $\Delta^{199}$Hg of $-1.44 \pm 0.12$‰ (2σ, n = 25) and $0.07 \pm 0.10$‰ (2σ, $n = 25$), respectively. Hg isotopic signatures in procedural standards are reported for $\delta^{202}$Hg (maximum 2σ = 0.17‰), $\Delta^{199}$Hg (maximum 2σ = 0.13‰), $\Delta^{200}$Hg (maximum 2σ = 0.08‰), $\Delta^{201}$Hg (maximum 2σ = 0.19‰) and $\Delta^{204}$Hg (maximum 2σ = 0.44‰).

## Stable isotope data analysis

We use $\Delta^{200}$Hg in NH remote $Hg^0$ and rainfall $Hg^{II}$ to quantify the atmospheric Hg deposition pathways to peat (Eqs. 4 and 5)[3].

$$\Delta^{XXX}Hg_{peat} = \alpha \times \Delta^{XXX}Hg_{Hg^0} - \theta \times \Delta^{XXX}Hg_{Hg^{II}} \quad (4)$$

$$\alpha + \theta = 1 \quad (5)$$

Symbols α and θ represent the proportion of Hg$^0$ and rainfall Hg$^{II}$ deposition, respectively.

## Reporting summary

Further information on research design is available in the Nature Portfolio Reporting Summary linked to this article.

## Data availability

Data generated in this study are provided in both the Source Data files and the Supplementary Information. Source data are provided in this paper.

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

## Acknowledgements
This study is supported by a Swedish Science Foundation grant to K.B. (Dnr2018-04695) and a Swiss National Science Foundation Ambizione grant to M.J. (No. PZ00P2_174101). We acknowledge ICOS Sweden and SITES for the provisioning of facilities, experimental support, and data. ICOS Sweden and SITES are funded by the Swedish Research Council as a national research infrastructure. We are grateful to Richard Bindler for his support with field equipment. We would like to thank Pernilla Löfvenius, Jacob Smeds, Xiangwen Zhang, Per Marklund, Rowan Messmer, Johan Westin, Paul Smith, Jutta Holst, Lamia Atouil, Matéo Wodiczko, Myriam Bupto, and Matthias Peichl for assistance in the field and/or data collection. A special thanks to Julie Brochet and Eloi Mathis for their great help with > 2000L peat water collection during the summer of 2021. We thank Manuela Fehr for her support with Hg stable isotope measurements at ETH Zurich, Switzerland. We are grateful to Chenyan Ma and the staff at Beamline 4B7A, BSRF for their assistance with the sulfur K-edge XANES spectroscopy measurements.

## Author contributions
K.B., M.J., M.N., S.O., W.Z., and C.L. designed the work. C.L., K.B., M.J., M.N., S.O., H.P., and W.Z. prepared and/or performed fieldwork. C.L., M.J., D.M., and Y.S. performed laboratory analyses. C.L. and K.B. led the discussion and paper writing. C.L., M.J., M.N., S.O., W.Z., D.M., U.S., M.E., H.P., Y.S., E.B., and K.B. contributed to data interpretation and writing.

## Funding

## Competing interests
The authors declare that they have no known competing financial interests or personal relationships that could have appeared to influence the work reported in this paper.
