## [Peer Review File · Nature Communications]

Mercury deposition and redox transformation processes in peatland constrained by mercury stable isotopesReviewer #1 (Remarks to the Author):

The manuscript is well written in general and provides a new evidence to show in details that how the peat soil gaseous Hg⁰ is derived from, by using both Hg concentration and Hg isotopic approaches. I enjoyed reading the manuscript although it is quite complicated, I found that the interpretation of Hg isotopic data is quite novel by considering d202, D199, and D200 simultaneously. I believed most of the interpretation by the authors to be correct, and I would like to provide a few comments or seek for clarifications:

L25-26: For the statement: "supersaturated gaseous Hg below the water table was likely created more by photoreduction of rainfall rather than release from peat soil.", I am not very clear about it, and is it possible to have such high levels of DGM all due to the surface photoreduction (likely limited to just a few cm with sunlight, right?)?

L61: For "while direct rainfall supply of HgII is not found to cause MDF and MIF", I wonder do you mean before or after binding to OM?

L64: How deep are you talking about peat surface? Is it possible for sunlight penetration to deeper layers? Has the team made any measurements in the field of light penetration? Because much of your conclusion is based on photochemical effects of Hg in the peat surface...

L67: Is it that S-Hg bonds are more prevalent in peat? How about other binding ligands?

L85: Regarding to the water table level, as the authors indicated later it is different in the wet season, I wonder how much variations of the water table level can occur over time? Does your findings apply to those wet periods?

L124-126: How different would be these two Sphagnum species in terms of binding Hg (i.e., dry deposition)? Has anyone measured or characterised their binding sites?

L147: Fig. 2: Very important and great dataset.

L176-178: I wonder can you add a figure of precipitation along with the D200Hg so we can see it clearly, perhaps in SI section?

L231: I suggest the authors state the levels observed in arctic tundra soil and forest soil in northern America. Was the later referred to the North America???

L326-327: How about the DOC levels in field vs. lab? How high are DOC levels in the peat groundwater?

L393: Fig. 6: Not sure if it is possible, how about adding a companion figure to indicate what may happen if there are trees in the peatland...?

L425-426: For "The dominant peat Hg loss caused by photoreduction can likely reduce Hg methylation rates by decreasing the HgII available for methylation", was levels of bioavailable HgII for methylation (or easily reducible HgII) shown explicitly?

Reviewer #2 (Remarks to the Author):

I was overall very impressed with the extensive data, some of it very novel, in this submission by Li et al. I think there are many important new things to learn using Hg stable isotopes, particularly in peatland systems. The main new things in this paper are the data about Hg isotopes in the peat pore space both above and below the water table. Overall, I think the writing in the paper is quite good, but I would say that it stops a little short of being overly persuasive in some parts. There are a number of clarifications I would like to see in the paper and I have at least one major thing that I think very much needs to be discussed, but beyond this, I think the paper will be a very nice addition to the literature about Hg cycling in peatland systems.

The one major thing is that it is not currently clear in the paper that the authors have considered the very large differences in Hg uptake that likely occur in vascular plants (almost every publication to date related to Hg isotopes) and Sphagnum moss, which dominates most peatland systems, including this one. Sphagnum is non-vascular with quite different "pseudostomata", so the uptake of Hg into moss is likely going to be quite different than it is for leaves on vascular plants. I hesitate to say that we know nearly enough about this process to rely on thinking, for example, that isotope fractionation effects and a proper interpretation of Hg uptake and storage can be reasonably accounted for without considering this. I suspect quite a bit of the Hg "uptake" in Sphagnum might be surface absorption. The authors consistently refer to this as "uptake", so some additional consideration in the paper is needed.

In addition to this, here are some specific comments on different parts of the paper:

- First line of abstract – is Hg uptake actually "constant"?
- Abstract and elsewhere – why is Methyl in Methyl-Hg capitalized?
- Line 47-50: Just "reported"? Not very specific for an introduction. Next sentence as well – it is much better not to necessarily explain what people have done, but rather to explain the major findings.
- Line 52-53: Repetitive to lines 45-46.
- Line 61: Is there documentation of whether Hg(II) absorption to peat causes fractionation?
- Line 76-78: Something about wording in this sentence is off. I think "Given..." to start is the problem.
- Section at line 110: The "associated controlling factors" in this section are quite unconvincing in my opinion. Possible, but nothing particularly convincing in here so some tone change or additional explanation of limitations may be necessary.
- Line 118-120: This sentence is unclear to me. Do you mean by modelling that this can be achieved or by inference of the process?
- Line 129: Are you assuming that productivity in Sphagnum and Hg uptake are definitely linked? Do we know this?
- Line 166: I find it odd that the authors use more global averages instead of data they have for local atmosphere. Why not use the atmospheric values you have from Degero? Seems an odd handling of these data, without adequate explanation of the choice.
- Line 172: More information about vegetation in this system is needed, particularly coverage of different species. I would guess that leaf uptake by some of the vascular shrubs and other plants (and then deposition) would differ from how Hg ends up in Sphagnum. Perhaps not for cap-200 Hg though?
- Line 178: Also likely changes in the magnitude of Hg wet deposition in these periods?
- Line 188 ("calculated" peat $\Delta^{200}\text{Hg}$): From the above section, is cap-200 Hg not directly measured? How exactly and why are you calculating it based on end members?
- Line 190-191: Wouldn't post-depositional photoreduction also have an impact on the cap-199 Hg? It does in Figure 1. This is invoked in the coming sentences, but if you include photoreduction as part of "mobility", that is a mistake.
- Line 315: Broadly groundwater or just surficial groundwater? Doesn't it seem implausible that DGM would be maintained from rain water for long periods of time?
- Line 323: Sphagnum is in fact quite amazing at holding quite a lot of water nearer the surface. This hours to days residence time seems potentially even a major underestimation and I see no citation for this statement. Beyond this though, I found

this paragraph to be the most interesting one of the paper. Very nice.

- Figure 6 mechanism in the unsaturated zone: This mechanism is not very clear to me as written. Maybe clarify that "air" is atmospheric and that it is moving into the unsaturated pore space? The mediation by dark abiotic reduction is a little unclear from the figure alone. Additionally, the percentages on the surface fluxes - I can easily infer 36 vs 64%, but for photoreduction, it is 27% of what?
- Line 408-409: I find statements like these overly vague for a conclusions section. Specifically what new information you are referring to? Is the above, some of it cited from other studies "new information" what you mean?
- Line 414-415: Do you feel for sure that this never happens? For example, I don't see that your measurements necessarily have specificity to the diurnal dynamics we know to occur throughout a day (several chamber-based studies, including in peatlands, on this). Maybe just more specificity is needed here if you are suggesting that only the very surface is where everything is happening.
- Line 495: Continuously for the whole summer?

Reviewer #3 (Remarks to the Author):

The manuscript presents results from field samples collected from an open boreal peat land system with the goal to better understand the fate of Hg in peatland and the export of methyl-mercury from peatland to downstream ecosystems. Mercury (Hg) stable isotope ratio measurements were used to investigate the influence of post depositional Hg transformation processes. Re-emission of Hg from the peatland after photoreduction of Hg(II) and dark abiotic oxidation of Hg(0) were identified as predominant processes after deposition of Hg from the atmosphere or via rainfall. Based on the isotope ratios, dissolved gaseous Hg in groundwater was described to result from reduction of Hg(II) in rainfall rather than reduction of Hg associated with peat. Post depositional Hg transformation processes are of interest because they can have an effect on the availability of Hg for methylation and because they may alter the Hg concentrations archived in peat and therefore affect their suitability as long term archive of atmospheric Hg (e.g. by non-quantitative retention and loss of mercury during peat diagenesis). This study concludes that there is a remobilization of 30% of deposited Hg and the controlling process is photoreduction. No other processes were identified based on the isotope composition of Hg in peat and that peat can be considered as suitable archive for Hg accumulation and atmospheric deposition.

The manuscript is well-written and clearly structured, and my overall impression of the manuscript is positive. The results are presented in clear figures and the used methods were mostly evaluated using common quality assurance. However, I do have some critical comments and requests for clarification about the methods and interpretations presented in the submitted manuscript. In summary, I believe the requests can be clarified and that a revised version of the manuscript addressing the following comments will probably be suitable for publication in Nature Communications.

Introduction:

The introduction presents the relevance of the topic, summarizes current literature and introduces the research questions. This section is well written and I have only some minor comments:

L 40: I get what you mean with "oxidized Hg(0)" (Hg which originates from the atmosphere as Hg(0) and is subsequently oxidized). But I think this is a bit confusing as Hg(0) is the reduced form and oxidized Hg(0) would be Hg(II). Consider rephrasing.

L 46: long term Hg sink

L 53-53: While I agree that it is of great importance to determine whether peatlands are net producers and exporters of MeHg, this is not a question addressed in the current

study. The study does not include measurements of MeHg or evaluate possible isotope fractionation caused by methylation. Methylation could also be considered a post-depositional Hg transformation process and the title of the manuscript as well as this sentence could suggest that this is part of the study. I suggest adding a sentence to clarify how the processes described in the next section are related to methylation and what conclusions about MeHg export can be made from these results.

L 66-67: If MIF caused by photoreduction can go in either direction depending on the Hg bonds, could both happen in parallel resulting in a mixed signal? What would this mean for process tracing?

Results and discussion

L106-107: What factors control whether Hg(0) diffuses to the atmosphere or into deeper peat layers? How much of the Hg(0) remains associated with the peat? Is there any analysis of Hg(0) in the peat solid material?

L. 107-108: Dark abiotic oxidation results in Hg(II) that is enriched in heavier isotopes compared to Hg(0), "...resulting in Hg(II) with (+)MDF and (-)MIF"

L 110-115: Please explain how the Hg accumulation rates were calculated in the methods section. How is peat decomposition accounted for in this? Or losses of Hg after deposition?

L 111-112: Please add that net accumulation rates reflected by present day Hg concentrations also include potential loss terms. This is important especially also in the context of loss through photoreduction discussed later on or export via streamflow.

L 117-118: Consider rephrasing this sentence, explaining that differences in HgAR can be explained by higher HgAR is confusing. Here it becomes important that the authors specify that the net accumulation consists of an uptake/accumulation term and a loss term. Both terms can be the reason for the observed differences in net HgAR.

L 118-119: The next sentence also needs explanation in terms of causality: "A greater HgAR can be achieved by higher peat Hg concentrations and/or chronology-based peat AR". In the process of calculating HgAR, higher Hg concentrations in the samples result in a higher HgAR. But is the process not in the other direction and the higher HgAR would result in higher observed peat Hg concentrations?

L 132-134: Interesting that the species with higher primary production also have higher Hg concentrations (per mass produced, more Hg is assimilated) and that the Hg is not "diluted" by the larger biomass accumulation in Hummock or concentrated by faster peat decomposition in Lawn. In that sense, would a higher peat AR not rather indicate the opposite trend? Alternative explanations are discussed in the following sections (higher Hg assimilation by different plant species, differences in Hg loss).

Figure 2: c) and d) axis label should state it's Hg concentrations.

L 165-166: The last part of this sentence should be removed or rephrased. Mixing approaches are usually per se based on the assumption that the mixing is conservative (otherwise it's transformation processes). If there are Hg sources involved that exhibit even-mass MIF, the result of the mixing will also mix these MIF signatures, resulting in a different $\Delta 200\text{Hg}$ value. In that sense, source mixing can alter even-mass MIF. (which is what you do in the next sentence by mixing rainfall and GEM endmembers).

L 170-171: Please add a reference to Eq. 3 and 4 for the formulas of the end member mixing.

L 174: It could be helpful for the reader to visually indicate the 1950-2000 and 1800-1950 periods in Figure 2 within the orange shading (dashed line or similar).

L 175-176 The differences in $\Delta 200\text{Hg}$ are subtle and based on few samples. Please indicate $n = xy$ so that this becomes apparent.

L 180: Terminology: " $\Delta 199\text{Hg}$ profiles exhibit overall increases" => shifts to more positive / less negative values. The delta (difference / deviation from NIST, which can be in either direction, positive or negative) becomes smaller.

L 183-184 Additional to changes in Hg emissions, the difference in relative contribution, as stated in lines 177-178 can also result from increased precipitation. Up until here it has not been established that the rainfall and GEM endmembers have different $\Delta 199\text{Hg}$ values, which could be important for the broader audience.

L 185: "the assumed conservative $\Delta 200\text{Hg}$ -derived wet deposition" is confusing. I get what you mean, but this should be written more accurately. The wet deposition is not $\Delta 200\text{Hg}$ -derived (and also not conservative), but the contribution of wet deposition to peat Hg is estimated using conservative endmember mixing based on $\Delta 200\text{Hg}$.

L 187-188: This sentence needs clarification because as it is, it's comparing $\Delta 199\text{Hg}$ values to $\Delta 200\text{Hg}$ values. It is unclear what you are comparing the measured $\Delta 199\text{Hg}$ values to. Do you mean the $\Delta 199\text{Hg}$ calculated from the mixing of atmospheric endmembers (based on the contributions derived from $\Delta 200\text{Hg}$), resulting in $\Delta 199\text{Hg}$ values which are less negative than the measured values? If so, please also refer to Eq. 3 and 4 and specify that this has been calculated using the same formula. How large is the difference between the calculated $\Delta 199\text{Hg}$ (based on source contributions derived from mixing using $\Delta 200\text{Hg}$) and the measured values? Can you estimate a uncertainty for the calculated values? How does this compare to analytical uncertainty?

L 189: The "red line" in Figure 3 is a York regression based on the individual data points and accounting for error in x and y direction (I assume). This is not the mixing line of two endmembers. While both might result in similar lines, this is not the same thing. Instead of the York regression I suggest adding the actual values you defined as endmembers (incl uncertainty) to Fig. 3 and a mixing line. Besides mixing of rainfall and GEM endmembers, is there a conceptual reason for a relationship between $\Delta 199\text{Hg}$ and $\Delta 200\text{Hg}$?

L 194-195: Lower $\Delta 199\text{Hg}$ => more negative $\Delta 199\text{Hg}$

L 197-200: This is a pity. The $\Delta 199\text{Hg}/\Delta 201\text{Hg}$ slope can be a very powerful indicator to differentiate photochemical reduction from abiotic reduction pathways.

L 203-205: Is there $\text{Hg}(0)$ associated with the solid phase?

L 214: Please add the references, "refs see text" is not helpful

L 219: The shift of the measured data compared to atmospheric $\text{Hg}(0)$ is rather small. I'm not convinced that this is sufficient to call it a $\Delta 199\text{Hg}$ anomaly (L 252-253)

L 240-242: Is this further supported by the peat soil Hg isotope data? Previously, the $\Delta 199\text{Hg}$ values in peat soil were explained by mixing of rainfall and GEM plus post depositional reduction of $\text{Hg}(\text{II})$. Does the oxidation of soil gas $\text{Hg}(0)$ fit in the story? If the soil gas $\text{Hg}(0)$ resulted from reduction of $\text{Hg}(\text{II})$ in peat, the soil gas $\text{Hg}(0)$ would likely also be isotopically lighter than atmospheric $\text{Hg}(0)$?

L 246: I understand that data on a potential convective flow is not considered. But would a diffusive flux of Hg be able to happen if an opposite convective flow existed? Is there any data on convective flow or any literature data to compare this with?

Figure 4: What is meant with "regression of a Rayleigh model" (L 258)? You had a

starting point (please define) and used the enrichment factors reported in Zheng et al., 2019 for dark abiotic oxidation of Hg(II) to calculate a Rayleigh model? But that is not a regression. Please provide more information about how this was calculated.

What is the reason for using a Rayleigh model for MDF in panel b)? Zheng et al. 2019 (Ref. 19) report enrichment factors for equilibrium fractionation. Additionally, they identified a larger isotope effect during initial stages of their experiments, which was attributed to kinetic effects. However, the enrichment factors apply for equilibrium fractionation. Either you assume there is isotope exchange between Hg(II) and Hg(0) and you apply an equilibrium model (linear) or you assume that isotope exchange is hindered and kinetic effects dominate the system. In the latter case a Rayleigh model could be applied but this would need better justification.

Visually, the data in panel b) could also be fitted with a linear trend line that would probably result in a similar goodness of fit?

What is the solid line with the red shading in b)?

The trajectory in c) is based on a starting point and a slope I assume? Could you specify this?

L 270: I think the order should be changed? The Hg(0) saturation level in the groundwater is 1500% and 4300% relative to peat soil gas phase and the atmosphere, respectively. Peat soil gas has lower Hg concentrations than atmosphere.

L 265-285: If the reduction rate of Hg(II) is slower than Hg(0) oxidation because of the high affinity of Hg(II) to NOM binding sites and there is no diffusion of Hg(0) from soil air to groundwater (supported by $\Delta 200\text{Hg}$, section below), what is the source of DGM in groundwater?

L 298-300: Please clarify what "its groundwater" means in line 300?

L 306-307: Neither biotic or abiotic reduction can explain why DGM $\Delta 200\text{Hg}$ values lie between rainfall and peat soil values (line 298). Either you consider $\Delta 200\text{Hg}$ to be within error for rainfall and re-write sentence starting in line 298. Otherwise, an explanation for the shift in $\Delta 200\text{Hg}$ is needed.

L 317: "...by dark chemical or microbial reduction, or by photochemistry." I think it's enough to just write by reduction and delete this last part. It's unspecific and those are pretty much all possible pathways.

L 317-319: This is not only in line with microbial reduction, all reduction pathways lead to Hg(0) product that is enriched in light isotopes.

L 319-321: Terminology: Higher and lower delta is not precise (s. comment above)

L 327-329: What was the criterion to use this slope, but not for the other $\Delta 201\text{Hg}$ data? Same for Table S2, some samples were deemed as unreliable (footnote), what is the criterion? You write that this is based on a generally significant relationship between $\Delta 201\text{Hg}$ and $\Delta 199\text{Hg}$ under the same mechanism. This should be written more clearly. The relationship between $\Delta 201\text{Hg}$ and $\Delta 199\text{Hg}$ can vary quite a bit depending on the process. Do you mean it has a different sign (one is negative, one positive?)

Figure 5: What does the grey area represent in c)

Inset d) is very small and hard to read. I wonder if it is even necessary?

L 359: low $\Delta 199\text{Hg}$ => negative $\Delta 199\text{Hg}$

L 366-368: Please also mention the uncertainties and limitations of the identification of processes based on the observed subtle differences in $\Delta 199\text{Hg}$. The differences between the observed Hg isotope ratios in peat Hg and atmospheric Hg are small (Figure 3).

L 370: Ref 73 reports results from experiments investigating the Hg isotope

fractionation during the volatilization of dissolved Hg(0) to the gas phase. This does not include fractionation from the reduction process itself. This is different from photoreduction or the emission from vegetation.

L 369-374: This needs more explanation. How exactly were these values calculated? What is the starting point for this calculation (i.e. how is the isotope value for f=0 determined)? How can you calculate the loss since 1800CE if the isotope ratios of atmospheric Hg or rainfall Hg isotope composition in 1800 is unknown. Not sure how well these results go with the conclusion that Hg is immobilized in the peat and not subject to any mechanisms redistributing Hg after deposition (lines 418-420).

L 375-381: If there is a yearly discharge of Hg complexed to organic matter, in my opinion this would represent a redistribution of Hg from peat to the liquid phase, probably also during the decomposition of the peat (it's originating from the peatland, L 377). Or do you assume that this discharge is mainly Hg derived from rainfall? If so, can this be supported by the difference in Hg isotope ratios (e.g. $\Delta^{200}\text{Hg}$)?

L 400: What is rHg(II)?

L 402: How would the fact that Hg(II) deposited from rainfall is more susceptible to photoreduction affect the mixing calculation based on $\Delta^{200}\text{Hg}$? Since one of the endmembers is affected more by the post depositional transformation process, the mixing calculation might not represent the actual contributions?

L 414-415: To what depth do you expect the photoreduction to happen? The Hg(0) produced from photoreduction would be released to the atmosphere directly and not end in peat soil gas, right?

L 417-418: It should be specified that this photoreduction of rainfall Hg(II) partially happens before deposition, and partially after.

L 425-426: This statement is a bit out of context because you did not really discuss methylation rates previously. Reduction of Hg(II) to Hg(0) would reduce the methylation rate compared to what scenario? Plus, this would be the same for any other reduction pathway and is not specific to photoreduction.

L 426-428: You calculate an approximate Hg loss of 30% by photoreduction, export via streamflow is less half of this (L 380), <15%. Overall, this is a loss of approx. one third of the deposited Hg. This was confusing at first, how can you identify post-depositional Hg transformation processes resulting in a loss of one third of Hg and at the same time conclude that this is small enough to consider peat a reliable archive? I think the key message is in lines 418-420 and it should be stated more clearly that your results indicate that this loss primarily affects Hg deposited by rainfall and not Hg incorporated in peat. Plus, your calculations of rates of Hg loss by photoreduction indicate that this loss is consistent over time.

Because this manuscript has a focus on using Hg isotope ratios, this last sentence remains a bit unclear to me whether you mean an archive for atmospheric Hg deposition or also the atmospheric Hg isotope ratios (would the Hg isotope ratios also be conserved in the peat associated Hg?).

Methods:

The presented data is of sufficient quality, despite the higher analytical uncertainty in the MIF data. The authors mostly used appropriate quality controls. However there are some questions regarding the methods that need clarification. Further, the calculations of accumulation rates or loss rates are not well explained and it is not entirely possible to follow all the steps or even replicate the calculations. The authors should provide a more detailed description of the calculation steps and the input parameters used. The effort needed to obtain Hg isotope data from peat groundwater is remarkable. However, it should also be noted that the dataset is therefore also limited and that any

interpretations should be made with caution.

L 454: Why is cleaning of peat slices with MilliQ water necessary?

XANES results are reported in the SI and only referred to in the main text without stating that these are the results of XANES analysis and without reference to the SI. This should be more clearly mentioned and referenced.

The sampling and pre-concentration for the groundwater samples needs some additional explanations. In order to sample 2000 L of water this needed approx. 100 bottles? How were the samples stored in between? The pre-concentration needs some specification how this large volume was handled (Fig. S13).

How was the recovery (yield) calculated? Was an aliquot of the peat groundwater sampled for isotope analysis also used to also analyze total Hg concentration? The methods only specify how DGM was measured in groundwater (Fig. S14). I assume this was the same setup for rainfall DGM analysis?

L 525-530: Rainfall DGM results should be reported in the SI along with the other results. Is there any control on how much Hg would be lost from purging a Hg(II) solution with similarly low concentrations? How were total Hg concentrations in rainfall determined?

L 533: I assume the four measurements are the reported DGM values. It does not become apparent in the main text that the results for groundwater DGM are aliquots of the same sampling. The effort of obtaining this data is huge and should be honored, but nonetheless this should be made transparent.

L 535: Fig. S3 shows a calibrated age-depth model. You mean Fig. S13?

L 536: Were multiple of the setups depicted in Fig. 13 run in parallel? In order to process 2000 L of water this probably needed several days of work? "Within 1h" becomes much less meaningful and I wonder how the remaining water was stored?

L 547: The "150ml rainfall and peat groundwater samples" are the ones in L 529-530? This is a bit confusing after the pre-concentration section describing other groundwater samples, plus the 150ml are not mentioned before and can't be used to identify which samples are meant.

L 561: Were the Fluka standards matrix matched? 20% acid strength is rather on the upper limit for efficient reduction in the CV system.

L 563: Standard-sample bracketing can be used to minimize effects of signal drifts during the measurement, however it cannot correct instrumental mass bias. Established methods for Hg isotope analysis using MC-ICP-MS usually involve both standard-sample bracketing and instrumental mass bias correction with thallium doping. The cited reference (Ref. 88) suggests: "We strongly suggest first correcting instrumental mass bias by simultaneous measurement of the certified thallium isotope standard (NIST SRM 997; 205Tl / 203Tl ratio of 2.38714) followed by sample-standard bracketing with the NIST SRM 3133 Hg standard." Is there a reason there was no Tl doping used in this study?

The method section should also include details about how the different Rayleigh models displayed in the Figures 4 and used to calculate photoreductive Hg loss were implemented. The calculation of the reductive loss, especially for older periods is not explained and should definitely be added. Further, the calculation of HgAR is not explained and details should be added.

Supporting information:

Figure S1: d) In order to sample 2000 L of water multiple such bottles must have been sampled? This must have taken quite long. How were the individual bottles stored? Was there a difference in Hg concentration (or other parameters for that matter) between the initial sampling and the last sampling?

Figure S11: see comment above, please add a description of the method approach used to calculate the loss rates.

Table S6: How was the yield calculated? "Hg recovered from trap and sampling yield" is not very clear. "Hg recovered from trap": Did you measure Hg(0) concentration before and then calculate the recovery after purging it onto the traps? What is meant with "sampling yield"?

Table S7: The 2SD for $\Delta^{201}\text{Hg}$ in Fig. S8 looks larger than the reported 0.16 ‰? Please verify that it is the same value.

Table S2 and S5: There is a relationship between $\Delta^{201}\text{Hg}$ and $\Delta^{199}\text{Hg}$ and generally these two values should follow similar trends. However, depending on the process involved (MIE, NVE) the relationship can be different. What was the exact criterion for the exclusion of certain $\Delta^{201}\text{Hg}$ results? This explanation should be further clarified.

Many thanks for your detailed and helpful comments! All the “updated line numbers” mentioned below refer to the revised version with “no mark-up”. References mentioned in response to three reviewers’ comments are listed at the end of this file. Please see below our point-by-point response in blue.

REVIEWER COMMENTS

Reviewer #1 (Remarks to the Author):

The manuscript is well written in general and provides a new evidence to show in details that how the peat soil gaseous Hg⁰ is derived from, by using both Hg concentration and Hg isotopic approaches. I enjoyed reading the manuscript although it is quite complicated, I found that the interpretation of Hg isotopic data is quite novel by considering d202, D199, and D200 simultaneously. I believed most of the interpretation by the authors to be correct, and I would like to provide a few comments or seek for clarifications:

We thank very much the reviewer for his/her recognition of our work and comments!

L25-26: For the statement: "supersaturated gaseous Hg below the water table was likely created more by photoreduction of rainfall rather than release from peat soil.", I am not very clear about it, and is it possible to have such high levels of DGM all due to the surface photoreduction (likely limited to just a few cm with sunlight, right?)?

Yes, it is possible. Indeed, this is a surprising result, but the isotopic evidence points clearly to photoreduction as the predominant process producing DGM in peat porewater, a process that is focused on just a few cm below the mire surface where sunlight can penetrate before being attenuated. This evidence is detailed in the manuscript at lines 259-313, Figures 5b and 5c (line 850). Briefly, Hg^{II} provided by rainfall and exposed to sunlight is likely an important source of DGM in peat groundwater. DGM in rainfall prior to deposition might also play a role in the peat groundwater DGM production (please see lines 281-285 and below).

“Notably the concentration of Hg⁰ in local rainfall ($27 \pm 5.7 \text{ pg L}^{-1}$, 1σ , $n = 6$, supplementary Table S7) is half of that in the surface peat groundwater DGM collected a few hours after the rain event ($54 \pm 1.8 \text{ pg L}^{-1}$, $n = 2$). Given equal water volumes of rainwater and groundwater, Hg in rainwater could at the most account for half of the quantity of Hg^{II} collected in the superficial peat groundwater on the same day.”

L61: For "while direct rainfall supply of Hg^{II} is not found to cause MDF and MIF", I wonder do you mean before or after binding to OM?

When we speak of “direct rainfall supply”, we mean before binding to OM. Details please see lines 814-815 (caption of Figure 1), “Precipitation also supplies Hg^{II} with non-significant isotopic fractionation expected when falling on the peatland”.

L64: How deep are you talking about peat surface? Is it possible for sunlight penetration to deeper layers? Has the team made any measurements in the field of light penetration? Because much of your conclusion is based on photochemical effects of Hg in the peat surface...

The peat surface generally refers to the top 3cm. This depth corresponds to the length of the green section of peat moss (Figures below. Left: hummock site; right: lawn site). The green section is where most of the photosynthesis occurs.

We have not made any measurements of light penetration in the field, but we rely on the color of the moss as an indicator of where there is light to drive photochemical reactions. To make it clear in the main text, phrase “based on the length of the green section of peat moss” has been added after “top 3cm” in lines 108-109.

L67: Is it that S-Hg bonds are more prevalent in peat? How about other binding ligands?

Yes, Hg-EXAFS experiments have shown that S-Hg bonds are prevalent in peat (Skylberg et al, 2006¹) since sulfur-compounds (e.g., thiols) have a stronger affinity to Hg than other binding ligands in peat, e.g., O-Hg and N-Hg (Skylberg, 2008²). For this reason, we have not explicitly considered other potential ligands. To make it clearer, we have added the phrase in line 66, “prevalent in peat^{27,28}” after “ligands with S-Hg bonds”.

L85: Regarding to the water table level, as the authors indicated later it is different in the wet season, I wonder how much variations of the water table level can occur over time? Does your findings apply to those wet periods?

The water table level fluctuates between 0 and –25cm relative to the peat surface (Peichl et al., 2013³). These water levels are measured continuously at the Degerö mire by the ICOS-Sweden research infrastructure (<https://www.icos-sweden.se/>). Some of our field sampling was conducted when the water table was near the surface. So we believe our findings apply to these wet periods as well.

L124-126: How different would be these two Sphagnum species in terms of binding Hg (i.e., dry deposition)? Has anyone measured or characterised their binding sites?

Great point! But unfortunately, to our knowledge, there are no published studies about the characterization of the Hg binding sites among these two Sphagnum species. This is beyond the

scope of this study and therefore we did not make any changes in the manuscript.

L147: Fig. 2: Very important and great dataset.

Thank you!

L176-178: I wonder can you add a figure of precipitation along with the D200Hg so we can see it clearly, perhaps in SI section?

Good idea! The newly made figure shown below and associated caption are now added in the supplementary Figure S7.

L231: I suggest the authors state the levels observed in arctic tundra soil and forest soil in northern America. Was the later referred to the North America???

Yes, we refer to data from North America (Sierra Nevada Mountains, California, U.S.). We have changed "Northern America" to "North America" in order to make it consistent in the text. We have revised the relevant text in lines 200-201: "*in line with reports from arctic tundra soil (1.06 ± 0.13 vs $0.54 \pm 0.14 \text{ ng m}^{-3}$)³⁴ and mineral forest soils in North America (e.g., 1.16 ± 0.35 vs $< 0.5 \text{ ng m}^{-3}$ below 20cm in Blodget Forest site)⁶¹*"

L326-327: How about the DOC levels in field vs. lab? How high are DOC levels in the peat groundwater?

The DOC levels in ground water are reported in Campeau et al., (2017) ⁴, (averaging $40.8 \pm 12 \text{ mg L}^{-1}$). Additionally, we also measured the DOC in the samples we worked with. They are consistent with the published data in Campeau et al., (2017) ⁴. Our average DOC concentration was $43.7 \pm 3.5 \text{ mg L}^{-1}$ ($n = 12$), ranging from 39.7 to 53.4 mg L^{-1} .

These DOC values analysed from our field samples are higher than those used in the lab experiment performed by Bergquist and Blum, (2007, 1 mg L^{-1}) ⁵. Our Hg/DOC ($0.33 - 1.61 \text{ ng/mg}$) is much lower than those used in their experiments ($34.6 - 8330 \text{ ng/mg}$ in Zheng and Hintelmann, (2009) ⁶; $60,000 - 100,000 \text{ ng/mg}$ in Bergquist and Blum, 2007). Zheng and Hintelmann, (2010) ⁷ suggests that Hg^0 in natural waters primarily originates from Hg^{II} bound to O/N donor groups in DOM. We therefore compared our trajectory line of $\delta^{202}\text{Hg}$ vs $\Delta^{199}\text{Hg}$ to the experimental trajectory describing photoreduction of Hg^{II} dissolved in the aqueous phase in the presence of DOC (Figure 5c).

L393: Fig. 6: Not sure if it is possible, how about adding a companion figure to indicate what may happen if there are trees in the peatland...?

Thank you for your interesting suggestion! The appearance of trees on a peatland has profound effects on the peat biogeochemistry, but considering this is outside the scope of our study we would like to kindly decline this suggestion.

L425-426: For "The dominant peat Hg loss caused by photoreduction can likely reduce Hg methylation rates by decreasing the Hg^{II} available for methylation", was levels of bioavailable Hg^{II} for methylation (or easily reducible Hg^{II}) shown explicitly?

No. The investigation of the levels of bioavailable Hg^{II} for methylation is beyond the scope of this study. The range of inorganic Hg, total Hg and methyl-Hg observed in the porewater at this mire have, however, been reported earlier by studies that we now include when describing what is known about Hg in this mire (lines 406-407. e.g., Bergman et al., 2012 ⁸; Fritsche et al., 2014 ⁹; Åkerblom et al., 2020 ¹⁰).

Reviewer #2 (Remarks to the Author):

I was overall very impressed with the extensive data, some of it very novel, in this submission by Li et al. I think there are many important new things to learn using Hg stable isotopes, particularly in peatland systems. The main new things in this paper are the data about Hg isotopes in the peat pore space both above and below the water table. Overall, I think the writing in the paper is quite good, but I would say that it stops a little short of being overly persuasive in some parts. There are a number of clarifications I would like to see in the paper and I have at least one major thing that I think very much needs to be discussed, but beyond this, I think the paper will be a very nice addition to the literature about Hg cycling in peatland systems.

We are very grateful to the reviewer for both his/her recognition of our work and for the specific comments!

The one major thing is that it is not currently clear in the paper that the authors have considered the very large differences in Hg uptake that likely occur in vascular plants (almost every publication to date related to Hg isotopes) and Sphagnum moss, which dominates most peatland systems, including this one. Sphagnum is non-vascular with quite different "pseudostomata", so the uptake of Hg into moss is likely going to be quite different than it is for leaves on vascular plants. I hesitate to say that we know nearly enough about this process to rely on thinking, for example, that isotope fractionation effects and a proper interpretation of Hg uptake and storage can be reasonably accounted for without considering this. I suspect quite a bit of the Hg "uptake" in Sphagnum might be surface absorption. The authors consistently refer to this as "uptake", so some additional consideration in the paper is needed.

Thank you for this insightful comment! Indeed, the Sphagnum moss does not have stomata. We also agree that much less is known about how Hg gets into/onto Sphagnum compared to what is known about these processes in vascular plants. So when we chose the term "uptake" in this paper, we intended to include the possibility of absorption. We now added "*including the possibility of surface adsorption of Hg⁰*" to make this clear when we first use the term "uptake" in line 37.

In addition to this, here are some specific comments on different parts of the paper:

- First line of abstract – is Hg uptake actually "constant"?

Thank you very much for your question! It is likely that plant uptake of Hg is not constant. So we have deleted the word, "*constant*".

- Abstract and elsewhere – why is Methyl in Methyl-Hg capitalized?

Thank you for pointing it out! There is no need to capitalize M in methyl-Hg. Now it is revised in lines 18-19 and 35.

- Line 47-50: Just "reported"? Not very specific for an introduction. Next sentence as well – it is much better not to necessarily explain what people have done, but rather to explain the major findings.

We follow the reviewer's suggestion and have changed the sentences as follows (Lines 45-48), "*Net Hg⁰ evasion from an open (tree-less) boreal peatland was measured with a micrometeorological*

method over the course of one year ¹². Variability in Hg⁰ evasion rates along a thawing permafrost fen - palsa - bog gradient in the subarctic were related to different amounts of Hg stored in the peat ¹³".

- Line 52-53: Repetitive to lines 45-46.

Thank you! To avoid the repetition, we have revised the relevant text to, "*While reduction and subsequent evasion might reduce the Hg^{II} available for methylation, it also raises questions about peatlands as a long-term Hg sink ¹⁴, and the suitability of peatlands as archives of earlier Hg deposition ^{15,16}. Given the importance of peatlands in global, regional and local Hg cycles, it is crucial to understand Hg deposition and biogeochemical transformation processes in the peat. The post-depositional processes of Hg related to Hg reduction and oxidation could be resolved by analyzing the abundance and composition of Hg isotopes.*" (lines 48-54).

- Line 61: Is there documentation of whether Hg(II) absorption to peat causes fractionation?

We are not aware of any study that has explored this question.

- Line 76-78: Something about wording in this sentence is off. I think "Given..." to start is the problem.

Thank you for pointing it out! We have now replaced "Given" by "Despite".

- Section at line 110: The "associated controlling factors" in this section are quite unconvincing in my opinion. Possible, but nothing particularly convincing in here so some tone change or additional explanation of limitations may be necessary.

Indeed, "controlling factors" sounds rather strong. We replaced it by a more proper phrase "*potential influences*" (line 92).

- Line 118-120: This sentence is unclear to me. Do you mean by modelling that this can be achieved or by inference of the process?

No. Sorry for the confusion. Now the sentence has been rephrased to "*Enhanced Hg sequestration in $\mu\text{g m}^{-2} \text{yr}^{-1}$ can be achieved by higher peat Hg assimilation in ng g^{-1} and/or higher peat biomass production ($\text{g cm}^{-2} \text{yr}^{-1}$, Figure 2a, 2b, 2c, 2d; Supplementary Text S1; Table S1; Figure S2, S3).*" (lines 103-105).

- Line 129: Are you assuming that productivity in Sphagnum and Hg uptake are definitely linked? Do we know this?

We are not aware of evidence that would lead us to make a different assumption in this regard.

- Line 166: I find it odd that the authors use more global averages instead of data they have for local atmosphere. Why not use the atmospheric values you have from Degero? Seems an odd handling of these data, without adequate explanation of the choice.

Thanks for your question! We have used the local Hg⁰ isotope data here, but no local rainfall Hg^{II} isotope data. Rainfall data are collected by other colleagues for a separate publication. However, the local rainfall values are not significantly different from the global average ones (Dr. W. Zhu, personal communication).

- Line 172: More information about vegetation in this system is needed, particularly coverage of different species. I would guess that leaf uptake by some of the vascular shrubs and other plants (and then deposition) would differ from how Hg ends up in Sphagnum. Perhaps not for cap-200 Hg though?

Details about vegetation have been included in the supplementary Figure S4, S5 and S6. Briefly, we presented detailed downcore plant macrofossil profiles for both hummock and lawn sites, including vascular plants and *Sphagnum* (Figure S4 and S5). We have also presented net primary productivity of different moss species (Figure S6).

Sphagnum species have much higher Hg concentrations than vascular plants at the Degerö peatland (Details please supplementary Table S3), which suggests *Sphagnum* species have a stronger capacity than vascular plants to take up Hg. We agree that $\Delta^{200}\text{Hg}$ signatures of endmembers are likely not altered by the Hg deposition to *Sphagnum* or vascular plants.

- Line 178: Also likely changes in the magnitude of Hg wet deposition in these periods?

Yes, this is what we have surmised in the preceding sentence (Lines 176-177 of the original manuscript.).

- Line 188 (“calculated” peat $\Delta^{200}\text{Hg}$): From the above section, is cap-200 Hg not directly measured? How exactly and why are you calculating it based on end members?

Many thanks for pointing out this mistake! It should be calculated peat $\Delta^{199}\text{Hg}$ here and we have measured $\Delta^{200}\text{Hg}$. This has been corrected in line 166.

- Line 190-191: Wouldn't post-depositional photoreduction also have an impact on the cap-199 Hg? It does in Figure 1. This is invoked in the coming sentences, but if you include photoreduction as part of "mobility", that is a mistake.

Yes, post-depositional photoreduction has an impact on the cap-199 Hg, besides Hg sources (i.e. atmospheric Hg⁰ and rainfall Hg^{II}) and other post-depositional processes (e.g., dark biotic and abiotic reactions). Photoreduction as one of the post depositional processes occurs in the photo-zones, while other post-depositional processes are likely everywhere in the peat profiles.

Sorry that we mistakenly wrote “calculated peat $\Delta^{200}\text{Hg}$ ” in line 166, which should be “calculated peat $\Delta^{199}\text{Hg}$ ”. Hope this correction won't cause any more the confusion that photoreduction as part of “mobility” is a mistake.

- Line 315: Broadly groundwater or just surficial groundwater? Doesn't it seem implausible that DGM would be maintained from rain water for long periods of time?

It is superficial groundwater (top 50cm), not ground water in general. Since it is our measurements that we are referring to, we trust that it is clear that we are not generalizing about deeper groundwater.

We acknowledge that it sounds a bit implausible that DGM from rain water would be maintained in even superficial groundwater for long periods of time. But it is the implication from the isotopic data.

- Line 323: Sphagnum is in fact quite amazing at holding quite a lot of water nearer the surface. This hours to days residence time seems potentially even a major underestimation and I see no citation for this statement. Beyond this though, I found this paragraph to be the most interesting one of the paper. Very nice.

Thank you for your recognition of this paragraph! Now three citations have been added in line 297 (Nijp et al., 2014¹¹; McCarter et al., 2014¹²; Nijp et al., 2017¹³). The estimated residence times refer to periods when rainfall input is driven water downward through the decimeter or two of unsaturated peat assuming “piston flow”. We agree that on average, residence times can be longer when conditions are dry. We hope that the context of the discussion makes it clear that we are referring to the periods of downward water movement.

- Figure 6 mechanism in the unsaturated zone: This mechanism is not very clear to me as written. Maybe clarify that "air" is atmospheric and that it is moving into the unsaturated pore space? The mediation by dark abiotic reduction is a little unclear from the figure alone. Additionally, the percentages on the surface fluxes - I can easily infer 36 vs 64%, but for photoreduction, it is 27% of what?

Many thanks for your helpful question! Now the phrase “air Hg⁰” has been revised to “atmospheric Hg⁰”.

It is 27% of total annual Hg deposition. Relevant details have been added in the caption of Figure 6 (lines 865-873 and below).

Hg deposition and mobility in a boreal peatland since 1800CE

Figure 6. The new conceptual model of Hg deposition and mobility in a boreal peatland based on the findings of this study. Precipitation input of Hg^{II} and plant uptake of Hg⁰ account for ~36% and ~64% of total Hg deposition (6.2 vs 10.9 $\mu\text{g m}^{-2} \text{yr}^{-1}$), respectively. Atmospheric Hg⁰ diffuses into the

unsaturated zone of the peat, as driven by the process of dark abiotic oxidation, even though this flux is negligible as compared to input of Hg in rainfall and plant uptake. Photoreduction of Hg^{II} to Hg^0 and subsequent evasion at the peat surface corresponds to 27% of the total Hg deposition ($4.4 \mu\text{g m}^{-2} \text{yr}^{-1}$). In contrast, the dark reduction of Hg^{II} to Hg^0 in the saturated zone of the peat is likely not significant compared to the photoreduction process.

- Line 408-409: I find statements like these overly vague for a conclusions section. Specifically what new information you are referring to? Is the above, some of it cited from other studies "new information" what you mean?

Thank you! To make this sentence clearer, it has been rephrased to "our results on detailed post-depositional Hg redox processes in peat" (lines 368-369). The cited information from other studies is to highlight how the findings from this study improve the process understanding of Hg cycling in peatlands.

- Line 414-415: Do you feel for sure that this never happens? For example, I don't see that your measurements necessarily have specificity to the diurnal dynamics we know to occur throughout a day (several chamber-based studies, including in peatlands, on this). Maybe just more specificity is needed here if you are suggesting that only the very surface is where everything is happening.

Based on our study, we are confident to conclude that upward net diffusion of Hg^0 via the air-filled pore space of the peat to the atmosphere is unlikely. We have actually collected samples covering night-time and we still found lower Hg^0 concentrations in the peatland air-filled sub-layers than in the atmosphere. For details on sampling time please see supplementary Table S4. Please note that evasion may be occurring through pathways that we have not measured, such as the saturated pores (water-filled).

Great idea to highlight in this paragraph that it is the peat surface where main Hg loss occurs! We added a sentence accordingly in lines 392-393, "*Our study highlights that it is the peat surface, instead of peat sub-layers, where main Hg loss occurs.*" Thank a lot for this suggestion to further improve this manuscript!

- Line 495: Continuously for the whole summer?

No, it is intermittently for the whole summer. The detailed sampling time please see supplementary Table S4.

Reviewer #3 (Remarks to the Author):

The manuscript presents results from field samples collected from an open boreal peat land system with the goal to better understand the fate of Hg in peatland and the export of methyl-mercury from peatland to downstream ecosystems. Mercury (Hg) stable isotope ratio measurements were used to investigate the influence of post depositional Hg transformation processes. Re-emission of Hg from the peatland after photoreduction of Hg(II) and dark abiotic oxidation of Hg(0) were identified as predominant processes after deposition of Hg from the atmosphere or via rainfall. Based on the isotope ratios, dissolved gaseous Hg in groundwater was described to result from reduction of Hg(II) in rainfall rather than reduction of Hg associated with peat. Post depositional Hg transformation processes are of interest because they can have an effect on the availability of Hg for methylation and because they may alter the Hg concentrations archived in peat and therefore affect their suitability as long term archive of atmospheric Hg (e.g. by non-quantitative retention and loss of mercury during peat diagenesis). This study concludes that there is a remobilization of 30% of deposited Hg and the controlling process is photoreduction. No other processes were identified based on the isotope composition of Hg in peat and that peat can be considered as suitable archive for Hg accumulation and atmospheric deposition.

The manuscript is well-written and clearly structured, and my overall impression of the manuscript is positive. The results are presented in clear figures and the used methods were mostly evaluated using common quality assurance. However, I do have some critical comments and requests for clarification about the methods and interpretations presented in the submitted manuscript. In summary, I believe the requests can be clarified and that a revised version of the manuscript addressing the following comments will probably be suitable for publication in Nature Communications.

We thank very much the reviewer for her/his recognition of our work as well as very helpful and detailed comments that we respond to below!

Introduction:

The introduction presents the relevance of the topic, summarizes current literature and introduces the research questions. This section is well written and I have only some minor comments:

L 40: I get what you mean with “oxidized Hg(0)” (Hg which originates from the atmosphere as Hg(0) and is subsequently oxidized). But I think this is a bit confusing as Hg(0) is the reduced form and oxidized Hg(0) would be Hg(II). Consider rephrasing.

Thank you! We agree that it is confusing. The phrase “oxidized Hg(0)” is now revised to “The oxidation of Hg⁰” (line 41).

L 46: long term Hg sink

Thank you. “Hg sink” is indeed better than “Hg store” here. It is now revised in updated line 50.

L 53-53: While I agree that it is of great importance to determine whether peatlands are net producers and exporters of MeHg, this is not a question addressed in the current study. The study does not include measurements of MeHg or evaluate possible isotope fractionation caused by methylation. Methylation could also be considered a post-depositional Hg transformation process

and the title of the manuscript as well as this sentence could suggest that this is part of the study. I suggest adding a sentence to clarify how the processes described in the next section are related to methylation and what conclusions about MeHg export can be made from these results.

We fully agree that we do not determine MeHg fluxes. Our focus is on the deposition and redox transformation processes as we now clarify in the revised title, "*Mercury deposition and redox transformation processes in peatland constrained by mercury stable isotopes*".

Methylation is also a post-depositional Hg process which could indeed be of relevance for the mobility of Hg. It is, however, difficult to apply the Hg isotope tool to reliably reconstruct the Hg loss by methylation because methylation and demethylation of Hg will only cause MDF, but peat vegetation uptake of Hg has already caused large and varying MDF (up to 2.5 ‰, Enrico et al., 2016¹⁴). We, however, again agree with the reviewer that we need to rephrase the sentences to in a clear way stating that this paper focuses on the potential transformation processes in peatlands that could be solved by Hg isotope fractionation (i.e. oxidation and reduction). Please see the revised lines 48-54 below.

"While reduction and subsequent evasion might reduce the Hg^{II} available for methylation, it also raises questions about peatlands as a long-term Hg sink¹⁴, and the suitability of peatlands as archives of earlier Hg deposition^{15,16}. Given the importance of peatlands in global, regional and local Hg cycles, it is crucial to understand Hg deposition and biogeochemical transformation processes in the peat. The post-depositional processes of Hg related to Hg reduction and oxidation could be resolved by analyzing the abundance and composition of Hg isotopes."

L 66-67: If MIF caused by photoreduction can go in either direction depending on the Hg bonds, could both happen in parallel resulting in a mixed signal? What would this mean for process tracing?
Results and discussion

Yes, it could happen in parallel. In general, the measured Hg isotope results may just give us the information of the dominant processes. For example, if both -O/-N and -S bonds are involved with photoreduction with the -S bond reaction dominates the process, then the MIF will likely show MIF under a Hg-S bond reaction.

L106-107: What factors control whether Hg(0) diffuses to the atmosphere or into deeper peat layers? How much of the Hg(0) remains associated with the peat? Is there any analysis of Hg(0) in the peat solid material?

Thank you for these interesting questions! We report the concentration gradients, and the constraints on the direction of movement. How much movement occurs is a multi-phase reactive transport in porous media, which is however beyond the scope of our paper.

We deem it likely that most of the Hg⁰ is oxidized by organic matter in peat, leaving a very small fraction of Hg⁰ attached to the peat. We think the findings reported by (Gilli et al., 2018)¹⁵ who made efforts to quantify soil gaseous Hg⁰, and found it to be below detection limit in Hg contaminated soils, supports this interpretation. While it would be interesting to measure the Hg⁰ adsorbed on the peat soil, measuring this would be a major challenge that we were not able to tackle in this study.

L 107-108: Dark abiotic oxidation results in Hg(II) that is enriched in heavier isotopes compared to Hg(0), "...resulting in Hg(II) with (+)MDF and (-)MIF"

Thank you for the correction! It is now revised in lines 822-823 (Figures and associated captions have been relocated to the end of the manuscript following the guidelines of the journal).

L 110-115: Please explain how the Hg accumulation rates were calculated in the methods section. How is peat decomposition accounted for in this? Or losses of Hg after deposition?

Thank you for pointing this out to make our manuscript clearer! The method to calculate Hg accumulation rates is now revised in lines 527-529. The calculated accumulation rates are the net fluxes, which include both primary biomass and decomposition processes. We focus on the discussion of net Hg flux and then potential Hg loss (e.g., lines 101-103; 118-120; 121-124).

L 111-112: Please add that net accumulation rates reflected by present day Hg concentrations also include potential loss terms. This is important especially also in the context of loss through photoreduction discussed later on or export via streamflow.

Good idea! The first part of this section have been revised to, "*Peatlands receive Hg mostly from atmospheric deposition through plant uptake of Hg⁰ and rainfall Hg^{II} supply. Post-depositional processes potentially result in a loss of Hg⁰ back to the atmosphere as a consequence of biotic and/or abiotic reduction of Hg^{II} to Hg⁰. Both deposition and post-depositional losses of Hg are reflected in the measured peat Hg accumulation rates (AR).*" (lines 93-96).

L 117-118: Consider rephrasing this sentence, explaining that differences in HgAR can be explained by higher HgAR is confusing. Here it becomes important that the authors specify that the net accumulation consists of an uptake/accumulation term and a loss term. Both terms can be the reason for the observed differences in net HgAR.

Thank you for this suggestion! We have replaced "*a great HgAR*" by "*a greater Hg sequestration*" to make it clear (line 102).

L 118-119: The next sentence also needs explanation in terms of causality: "A greater HgAR can be achieved by higher peat Hg concentrations and/or chronology-based peat AR". In the process of calculating HgAR, higher Hg concentrations in the samples result in a higher HgAR. But is the process not in the other direction and the higher HgAR would result in higher observed peat Hg concentrations?

Sorry for the confusion. It should be the higher Hg deposition and peat AR that result in higher HgAR. It is now rephrased to be clearer, "*Enhanced Hg sequestration in $\mu\text{g m}^{-2} \text{yr}^{-1}$ can be achieved by higher peat Hg assimilation in ng g^{-1} and/or higher peat biomass production ($\text{g cm}^{-2} \text{yr}^{-1}$, Figure 2a, 2b, 2c, 2d; Supplementary Text S1; Table S1; Figure S2, S3)*" (lines 103-105).

L 132-134: Interesting that the species with higher primary production also have higher Hg concentrations (per mass produced, more Hg is assimilated) and that the Hg is not "diluted" by the larger biomass accumulation in Hummock or concentrated by faster peat decomposition in Lawn. In that sense, would a higher peat AR not rather indicate the opposite trend? Alternative explanations are discussed in the following sections (higher Hg assimilation by different plant species, differences in Hg loss).

Hg won't be "diluted" by higher biomass production, as the reviewer also mentioned.

Yes, a higher peat AR could indicate a higher HgAR if the atmospheric Hg level is relatively constant and deposited Hg is largely mobile.

Figure 2: c) and d) axis label should state it's Hg concentrations.

Indeed. Thank you! It is revised now (line 825).

L 165-166: The last part of this sentence should be removed or rephrased. Mixing approaches are usually per se based on the assumption that the mixing is conservative (otherwise it's transformation processes). If there are Hg sources involved that exhibit even-mass MIF, the result of the mixing will also mix these MIF signatures, resulting in a different $\Delta^{200}\text{Hg}$ value. In that sense, source mixing can alter even-mass MIF. (which is what you do in the next sentence by mixing rainfall and GEM endmembers).

Thank you so much for this insightful suggestion! The latter part of the sentence, "*and source mixing (e.g., anthropogenic Hg input ^{56,57})*", is now deleted to be correct.

L 170-171: Please add a reference to Eq. 3 and 4 for the formulas of the end member mixing.

Thank you! Enrico et al., (2016) is now added to updated Eq.4 and 5 in revised manuscript at line 551.

L 174: It could be helpful for the reader to visually indicate the 1950-2000 and 1800-1950 periods in Figure 2 within the orange shading (dashed line or similar).

Thank you! But we would like to kindly decline this suggestion since 1/ there are dash lines to separate the 1950-2000 and 1800-1950 in Figure 2a & 2b, and 2/ the figures 2e-2j will look much busier if more dash lines are added within the orange shading.

L 175-176 The differences in $\Delta^{200}\text{Hg}$ are subtle and based on few samples. Please indicate n= xy so that this becomes apparent.

Good idea! Thank you! They are now added in lines 152-153.

L 180: Terminology: " $\Delta^{199}\text{Hg}$ profiles exhibit overall increases" => shifts to more positive / less negative values. The delta (difference / deviation from NIST, which can be in either direction, positive or negative) becomes smaller.

Agree. Thank you! It is now replaced by "*shifts to more positive values*" (line 158).

L 183-184 Additional to changes in Hg emissions, the difference in relative contribution, as stated in lines 177-178 can also result from increased precipitation. Up until here it has not been established that the rainfall and GEM endmembers have different $\Delta^{199}\text{Hg}$ values, which could be important for the broader audience.

Thank you for this helpful suggestion! The phrase, "*with distinct $\Delta^{199}\text{Hg}$ signatures*", is now added after "*by either a change in the relative contribution from dry and wet deposition*" in line 161.

L 185: "the assumed conservative $\Delta^{200}\text{Hg}$ -derived wet deposition" is confusing. I get what you

mean, but this should be written more accurately. The wet deposition is not $\Delta^{200}\text{Hg}$ -derived (and also not conservative), but the contribution of wet deposition to peat Hg is estimated using conservative endmember mixing based on $\Delta^{200}\text{Hg}$.

Thank you again for this helpful suggestion! Now the phrase has been revised to “*the assumed conservative $\Delta^{200}\text{Hg}$ -derived contribution of wet deposition*” to be more logical (lines 163-164).

L 187-188: This sentence needs clarification because as it is, it's comparing $\Delta^{199}\text{Hg}$ values to $\Delta^{200}\text{Hg}$ values. It is unclear what you are comparing the measured $\Delta^{199}\text{Hg}$ values to. Do you mean the $\Delta^{199}\text{Hg}$ calculated from the mixing of atmospheric endmembers (based on the contributions derived from $\Delta^{200}\text{Hg}$), resulting in $\Delta^{199}\text{Hg}$ values which are less negative than the measured values? If so, please also refer to Eq. 3 and 4 and specify that this has been calculated using the same formula. How large is the difference between the calculated $\Delta^{199}\text{Hg}$ (based on source contributions derived from mixing using $\Delta^{200}\text{Hg}$) and the measured values? Can you estimate a uncertainty for the calculated values? How does this compare to analytical uncertainty?

Sorry for the confusion. We mistakenly wrote $\Delta^{200}\text{Hg}$, instead of $\Delta^{199}\text{Hg}$, leading to the confusion of comparison between $\Delta^{199}\text{Hg}$ and $\Delta^{200}\text{Hg}$. Now $\Delta^{200}\text{Hg}$ is revised to $\Delta^{199}\text{Hg}$ in line 166, as also responded to one of the comments from review 2.

The difference between the calculated $\Delta^{199}\text{Hg}$ and the measured values ranges from -0.41‰ to -0.006‰ in lawn, and from -0.47‰ to -0.002‰ in hummock with an exceptional value of 0.16. The uncertainty of the calculated values estimated from Monte Carlo simulation varies little, from 0.119‰ to 0.123‰ (1σ), but it is higher than the analytical uncertainty of 0.065‰ (1σ , $2\sigma = 0.13\text{‰}$).

L 189: The “red line” in Figure 3 is a York regression based on the individual data points and accounting for error in x and y direction (I assume). This is not the mixing line of two endmembers. While both might result in similar lines, this is not the same thing. Instead of the York regression I suggest adding the actual values you defined as endmembers (incl uncertainty) to Fig. 3 and a mixing line. Besides mixing of rainfall and GEM endmembers, is there a conceptual reason for a relationship between $\Delta^{199}\text{Hg}$ and $\Delta^{200}\text{Hg}$?

We thank the reviewer very much for this comment and suggestion! Yes, the red mixing line represents the York regression based on the individual data points and associated uncertainties. We did also try the mixing lines between two atmospheric endmembers (mean and uncertainties). These two approaches indeed give very similar lines with similar slopes (2.99 vs 3.08, former vs latter) as the reviewer also mentioned. Despite this, we agree with the reviewer that a mixing model of two endmembers makes more sense. We have revised Figure 3 accordingly (Figure below and lines 832-841 of the manuscript).

Currently the processes of even/odd MIF in atmospheric end members are not yet well understood to derive a conceptual reason, even though the correlation can be quite significant (e.g., Jiskra et al., 2015¹⁶). Further investigations are needed to explore these processes.

L 194-195: Lower $\Delta^{199}\text{Hg}$ => more negative $\Delta^{199}\text{Hg}$

Thank you! It is now revised to “more negative $\Delta^{199}\text{Hg}$ ”.

L 197-200: This is a pity. The $\Delta^{199}\text{Hg}/\Delta^{201}\text{Hg}$ slope can be a very powerful indicator to differentiate photochemical reduction from abiotic reduction pathways.

Indeed, $\Delta^{199}\text{Hg}/\Delta^{201}\text{Hg}$ slope can be a powerful tool. But we need to be honest with our analytical quality of $\Delta^{201}\text{Hg}$ and try not to over-interpret it.

L 203-205: Is there $\text{Hg}(0)$ associated with the solid phase?

Yes, there is likely some Hg^0 associated with the solid phases, but that it is likely to be small, based on the findings reported by Gilli et al., (2018)¹⁵ who quantified soil Hg^0 , and found it to be below detection limit.”

L 214: Please add the references, “refs see text” is not helpful

Thank you! We have now replaced “refs see text” to “(3, 19, 48-56)” in the caption of Figure 3 (line 837).

L 219: The shift of the measured data compared to atmospheric $\text{Hg}(0)$ is rather small. I’m not convinced that this is sufficient to call it a $\Delta^{199}\text{Hg}$ anomaly (L 252-253)

Thank you for this comment! Indeed, the difference of $\Delta^{199}\text{Hg}$ between measured data and atmospheric Hg^0 is not large. But the shift we refer to here concerns the distance from peat $\Delta^{199}\text{Hg}$ to the mixing line as stated in lines 838-840 (length of purple dash line in Figure 3).

L 240-242: Is this further supported by the peat soil Hg isotope data? Previously, the $\Delta^{199}\text{Hg}$ values in peat soil were explained by mixing of rainfall and GEM plus post depositional reduction of Hg(II). Does the oxidation of soil gas Hg(0) fit in the story? If the soil gas Hg(0) resulted from reduction of Hg(II) in peat, the soil gas Hg(0) would likely also be isotopically lighter than atmospheric Hg(0)?

The downward diffusive flux caused by dark oxidation of Hg⁰ is negligible as compared to total Hg deposition, so the Hg isotope signatures of this diffusive flux is hardly visible in peat soil Hg isotope data.

Yes, if the soil gas Hg⁰ came from the reduction of Hg^{II} in peat, the soil gas Hg⁰ would likely be isotopically lighter than atmospheric Hg⁰ after plant preferential uptake of light Hg⁰ isotopes. So we think the interpretation in the manuscript does fit with the observed data in this study.

L 246: I understand that data on a potential convective flow is not considered. But would a diffusive flux of Hg be able to happen if an opposite convective flow existed? Is there any data on convective flow or any literature data to compare this with?

A diffusive flux of Hg could likely happen if an opposite convective flow existed. There are unfortunately no other available data to be compared with.

Figure 4: What is meant with "regression of a Rayleigh model" (L 258)? You had a starting point (please define) and used the enrichment factors reported in Zheng et al., 2019 for dark abiotic oxidation of Hg(II) to calculate a Rayleigh model? But that is not a regression. Please provide more information about how this was calculated.

What is the reason for using a Rayleigh model for MDF in panel b)? Zheng et al. 2019 (Ref. 19) report enrichment factors for equilibrium fractionation. Additionally, they identified a larger isotope effect during initial stages of their experiments, which was attributed to kinetic effects. However, the enrichment factors apply for equilibrium fractionation. Either you assume there is isotope exchange between Hg(II) and Hg(0) and you apply an equilibrium model (linear) or you assume that isotope exchange is hindered and kinetic effects dominate the system. In the latter case a Rayleigh model could be applied but this would need better justification.

Visually, the data in panel b) could also be fitted with a linear trend line that would probably result in a similar goodness of fit?

What is the solid line with the red shading in b)?

The trajectory in c) is based on a starting point and a slope I assume? Could you specify this?

Thank you so much for these brilliant comments! We actually felt thrilled when working on them! Indeed, it is improper to use the Rayleigh kinetic model in Figure 4b because 1/ lower Hg⁰ concentration in peat soil air than atmosphere indicates a net oxidation (dark abiotic) of Hg⁰ in the unsaturated zone, and 2/ the isotopic signatures of Hg in atmosphere and peat soil air match the equilibrium fractionation theory expected for a closed system. It may be argued that the atmosphere-peat soil (water unsaturated zone) is not a fully closed system, in which equilibrium isotope effects (EIE) would not be expected. However, the downward flux of Hg⁰ from the atmosphere is negligible as compared to total Hg deposition and likely of an episodic character (sometimes a flux sometimes not). Therefore, on a longer term, we believe it is very reasonable to approximate the system as semi-closed and to apply the model for EIE between Hg⁰ and Hg^{II} bonded to NOM-RS functional groups as the dominant process regulating Hg isotopic composition, in accordance with experimental results of Zheng et al., (2019)¹⁷. Even though we have a net oxidation of Hg⁰ in the sub-peat layer (water unsaturated zone) and over a longer time-period a net flux of Hg⁰ downwards from the atmosphere, this flux and the oxidation of Hg⁰ is so small that the kinetic

isotope effect of Hg^0 oxidation is overprinted by the EIE due to Hg^0 and $\text{Hg}(\text{NOM-RS})_2$ isotopic exchange.

We fitted a York regression line (equilibrium model) to our data of $\delta^{202}\text{Hg}$ vs Hg concentration in the updated Figure 4b below (lines 842-849), as the reviewer also suggested. Still, episodically when the peat unsaturated zone is not in an equilibrium condition with the atmosphere, diffusion of Hg^0 into the peat sub-surface likely occurs. In the absence of time-constrained data on concentration of Hg^0 in atmosphere and peat soil, we for simplicity calculated an average diffusive flux from the atmosphere to the peat sub-surface during the snow-free seasons. This estimated maximum diffusive flux is negligible as compared to the total Hg deposition (lines 221-231), so we can conclude with confidence that this flux will not contribute significantly to peatland Hg input. But the direction of the diffusive flux still highlights a net deposition of atmospheric Hg to the peat unsaturated zone, instead of a net Hg^0 evasion.

The solid line with the red shading in the original Figure 4b was the trajectory line from Zheng et al., (2019)¹⁷. Now we have updated the Figure 4b below with the York linear regression (as the reviewer also suggested). The trajectory line in original Figure 4c was based on the slope of $\Delta^{199}\text{Hg}$ vs $\delta^{202}\text{Hg}$ in Zheng et al., (2019)¹⁷ with the starting point based on our atmospheric Hg data. In the updated Figure 4c below, we decided to remove this improper trajectory line. Instead, we added the fractionation factors from Zheng et al., (2019)¹⁷ in lines 208-216 with more explanations.

L 270: I think the order should be changed? The Hg(0) saturation level in the groundwater is 1500% and 4300% relative to peat soil gas phase and the atmosphere, respectively. Peat soil gas has lower Hg concentrations than atmosphere.

Good idea to shift the order since we presented soil gas first and then atmosphere in the previous sentence. It is now revised in line 240. Thank you!

L 265-285: If the reduction rate of Hg(II) is slower than Hg(0) oxidation because of the high affinity of Hg(II) to NOM binding sites and there is no diffusion of Hg(0) from soil air to groundwater (supported by Δ²⁰⁰Hg, section below), what is the source of DGM in groundwater?

Regarding the source of DGM in groundwater, we have interpreted in lines 273-275, “The Δ²⁰⁰Hg-based mass balance derived from Monte Carlo simulations shows that 48% of this DGM originates from peat groundwater Hg^{II} (27% - 64%, IQR), and 52% from rainwater Hg^{II} (35% - 73%, IQR).”.

The part of DGM from peat groundwater likely sources from photoreduction, instead of the expected low dark reduction. We have added the word, “dark”, in front of “reduction of Hg^{II}” in line 293, to make this sentence clearer.

L 298-300: Please clarify what “its groundwater” means in line 300?

Thank you! It is now clarified to “*peat groundwater*” in line 273.

L 306-307: Neither biotic or abiotic reduction can explain why DGM $\Delta^{200}\text{Hg}$ values lie between rainfall and peat soil values (line 298). Either you consider $\Delta^{200}\text{Hg}$ to be within error for rainfall and re-write sentence starting in line 298. Otherwise, an explanation for the shift in $\Delta^{200}\text{Hg}$ is needed.

Thank you for pointing this out! It was $\delta^{202}\text{Hg}$ in “the trajectory of $\Delta^{200}\text{Hg}$ vs $\delta^{202}\text{Hg}$ ” that we would like to highlight. We have deleted “ $\Delta^{200}\text{Hg}$ vs” in this sentence to avoid the confusion about $\Delta^{200}\text{Hg}$ since this isotope signature is not significantly affected by both biotic and abiotic reductions.

L 317: “...by dark chemical or microbial reduction, or by photochemistry.” I think it’s enough to just write by reduction and delete this last part. It’s unspecific and those are pretty much all possible pathways.

Thank you! We agree with you! Now the sentence is revised to, “*While likely more than one half of the groundwater DGM can be explained by rainfall DGM, the rest might be attributed to reduction of rain water Hg^{II} after deposition to peat.*” In lines 288-290.

L 317-319: This is not only in line with microbial reduction, all reduction pathways lead to $\text{Hg}(0)$ product that is enriched in light isotopes.

We agree! The phrase is now revised to “is in line with microbial and abiotic reduction of rainwater Hg^{II} ” in line 291.

L 319-321: Terminology: Higher and lower delta is not precise (s. comment above)

Thank you! We have now replaced “*lower*” by “*more negative*”, and “*higher*” by “*more positive*” in lines 292 and 293-294, respectively.

L 327-329: What was the criterion to use this slope, but not for the other $\Delta^{201}\text{Hg}$ data? Same for Table S2, some samples were deemed as unreliable (footnote), what is the criterion? You write that this is based on a generally significant relationship between $\Delta^{201}\text{Hg}$ and $\Delta^{199}\text{Hg}$ under the same mechanism. This should be written more clearly. The relationship between $\Delta^{201}\text{Hg}$ and $\Delta^{199}\text{Hg}$ can vary quite a bit depending on the process. Do you mean it has a different sign (one is negative, one positive?)

Figure 5: What does the grey area represent in c) Inset d) is very small and hard to read. I wonder if it is even necessary?

Thanks for these comments! That we used the slope of $\Delta^{199}\text{Hg}$ vs $\Delta^{201}\text{Hg}$ in Figure 5d earlier was because the slope was mainly affected by the published rainfall data that we used from other literatures, not the $\Delta^{199}\text{Hg}$ vs $\Delta^{201}\text{Hg}$ in our four DGM samples. Now we have decided not to use any $\Delta^{201}\text{Hg}$ data from our study due to relatively high analytical uncertainties in the CRMs. So we deleted the sentence in original lines 327-329 and Figure 5d, which are actually not very necessary.

The grey area represents the rainfall Hg^{II} data, which are removed since the Figure 5d was no longer used. Footnotes in Table S2, Table S5, and Table S8 have been revised to highlight that we will not use $\Delta^{201}\text{Hg}$ data for any interpretation, but just presenting the data just in case other research groups want to compare their results with.

L 359: low $\Delta^{199}\text{Hg}$ => negative $\Delta^{199}\text{Hg}$

Thank you! It is revised now in lines 319-320.

L 366-368: Please also mention the uncertainties and limitations of the identification of processes based on the observed subtle differences in $\Delta^{199}\text{Hg}$. The differences between the observed Hg isotope ratios in peat Hg and atmospheric Hg are small (Figure 3).

Thank you for this suggestion! In the following three paragraphs, we present the uncertainties quantitatively. At just this particular line though, we would like to leave the text as it is since the interpretation here (lines 326-328) is qualitative.

L 370: Ref 73 reports results from experiments investigating the Hg isotope fractionation during the volatilization of dissolved Hg(0) to the gas phase. This does not include fractionation from the reduction process itself. This is different from photoreduction or the emission from vegetation.

Indeed, original ref 73 by Zheng et al., (2007)¹⁸ is not about photoreduction or emission from vegetation. We did not use their isotopic enrichment factors to calculate Hg reduction. Instead, we focus on the theory of “Rayleigh fractionation” that Zheng et al. have applied in their paper. We agree that this ref can cause confusion. So we decided to delete it and only keep the one by Criss, 1995 (Line 330). Thank you for the comment!

L 369-374: This needs more explanation. How exactly were these values calculated? What is the starting point for this calculation (i.e. how is the isotope value for $f=0$ determined)? How can you calculate the loss since 1800CE if the isotope ratios of atmospheric Hg or rainfall Hg isotope composition in 1800 is unknown. Not sure how well these results go with the conclusion that Hg is immobilized in the peat and not subject to any mechanisms redistributing Hg after deposition (lines 418-420).

Thank you for these comments! As partially mentioned in lines 329-331, we made use of the Rayleigh fractionation model (Criss, 1999) and the isotopic enrichment factors for photochemical reduction of Hg^{II} on foliage ($E^{199}\text{Hg}_{\text{reactant/product}} = 0.49$, Yuan et al., 2019¹⁹). The starting point of $\Delta^{199}\text{Hg}$ for the calculation is the calculated $\Delta^{199}\text{Hg}$ based on the mixing line of $\Delta^{199}\text{Hg}$ vs $\Delta^{200}\text{Hg}$ between two atmospheric end-members. We used the 1/ calculated $\Delta^{199}\text{Hg}$ as the reactant, 2/ the peat measured $\Delta^{199}\text{Hg}$ as the product, and 3/ $E^{199}\text{Hg}_{\text{reactant/product}}$ as the enrichment factor in the Rayleigh fractionation model. A new paragraph on the detailed calculation has been added in the supplementary Text S4.

The past Hg stable isotope composition in atmospheric end members remains unknown, which provides an inherent uncertainty in the use of these isotopes from archives. But to our general knowledge, Hg stable isotope composition in atmospheric end members is likely relatively constant since 1800CE based on the following reasons. 1/ The isotope box model by Sun et al., (2016)²⁰ suggested relatively stable atmospheric end member composition over time. 2/ A comprehensive review on historical Hg isotope signatures in sediments also suggests that sediment $\Delta^{200}\text{Hg}$ may be used to identify different sources of Hg deposition (Lee and Kwon et al., 2021)²¹.

The past trend of Hg deposition to peatlands is a mixing of atmospheric sources. If the atmospheric Hg isotope composition remains relatively constant, it is reasonable to use modern atmospheric Hg isotope signatures to interpret the past Hg deposition. Currently it is broadly accepted in the community to do so. Quite some studies on sediment and peat have successfully applied the modern

atmospheric Hg isotope composition as endmembers to reconstructing the past Hg deposition (e.g., Yin et al., 2016²²; Enrico et al., 2017²³; Kurz et al., 2019²⁴; Zerkle et al., 2020²⁵; Jiskra et al., 2022²⁶; Li et al., 2023²⁷).

L 375-381: If there is a yearly discharge of Hg complexed to organic matter, in my opinion this would represent a redistribution of Hg from peat to the liquid phase, probably also during the decomposition of the peat (it's originating from the peatland, L 377). Or do you assume that this discharge is mainly Hg derived from rainfall? If so, can this be supported by the difference in Hg isotope ratios (e.g. $\Delta^{200}\text{Hg}$)?

Thanks a lot for these insightful comments! We concur with you that the Hg in the streamflow from the mire is related to the decomposition of the peat that supplies the dissolved organic matter in the streamflow. This can be indirectly supported by Jiskra et al., (2017)²⁸, which shows that runoff of boreal forest soils has the same Hg isotope signatures as the ones in the soil. Furthermore, ¹⁴C dating of the OM in streamflow from the Degero mire puts it at less than half a century old (Campeau et al., 2017)⁴. To make this clearer, we replaced "*originating from the peatland*" to "*originating from the peatland system*" in line 337; and stated "*The ¹⁴C dating of the organic matter in streamflow from the Degerö mire shows that the organic matter is less than half a century old²⁶, suggesting that it is Hg deposited during the last 50 years that is possibly mobilized by water flow through the peatland system*" (lines 339-342).

L 400: What is rHg(II)?

It should be Hg^{II}. Many thanks for pointing this typo mistake.

L 402: How would the fact that Hg(II) deposited from rainfall is more susceptible to photoreduction affect the mixing calculation based on $\Delta^{200}\text{Hg}$? Since one of the endmembers is affected more by the post depositional transformation process, the mixing calculation might not represent the actual contributions?

Photoreduction of rainfall Hg^{II} would not affect the mixing calculation based on $\Delta^{200}\text{Hg}$ since $\Delta^{200}\text{Hg}$ would not be altered during all the post-depositional processes. But photoreduction can remove part of the rainfall Hg^{II} amount, leaving less Hg^{II} $\Delta^{200}\text{Hg}$ mixing with the Hg⁰ $\Delta^{200}\text{Hg}$ by plant uptake. This is not a problem for the calculation since we focus on the net flux contribution after the Hg loss during potential post-depositional transformation processes.

L 414-415: To what depth do you expect the photoreduction to happen? The Hg(0) produced from photoreduction would be released to the atmosphere directly and not end in peat soil gas, right?

We expect that the photoreduction happens primarily in the top 3cm where the moss are green, which is an indication of the depth to which photosynthesis occurs. The Hg⁰ produced from photoreduction could diffuse more easily to the atmosphere compared to the peat soil gas phase. But we cannot exclude some small diffusion of Hg into the peat soil gas following the concentration gradient of Hg between atmosphere and peat soil gas phase.

L 417-418: It should be specified that this photoreduction of rainfall Hg(II) partially happens before deposition, and partially after.

Thank you for this suggestion! The phrase, "*before and/or after deposition to peatland*", has been added to line 379 to make it clear.

L 425-426: This statement is a bit out of context because you did not really discuss methylation rates previously. Reduction of Hg(II) to Hg(0) would reduce the methylation rate compared to what scenario? Plus, this would be the same for any other reduction pathway and is not specific to photoreduction.

We agree that we do not discuss methylation rates in much detail. We mention methylation rates here to match up with the opening phrases in the Introduction. This is based on the assumption that the amount of Hg is a factor in the rate of methylation as stated in the original lines 425-426, *“The dominant peat Hg loss caused by photoreduction can likely reduce Hg methylation rates by decreasing the Hg^{II} available for methylation”*. To make this sentence more concise, we revised it to *“The dominant peat Hg loss caused by photoreduction can likely reduce Hg methylation rates by decreasing the size of this weakly-bond Hg^{II} pool which is expected to have a high availability for methylation.”* (lines 390-392).

L 426-428: You calculate an approximate Hg loss of 30% by photoreduction, export via streamflow is less half of this (L 380), <15%. Overall, this is a loss of approx. one third of the deposited Hg. This was confusing at first, how can you identify post-depositional Hg transformation processes resulting in a loss of one third of Hg and at the same time conclude that this is small enough to consider peat a reliable archive? I think the key message is in lines 418-420 and it should be stated more clearly that your results indicate that this loss primarily affects Hg deposited by rainfall and not Hg incorporated in peat. Plus, your calculations of rates of Hg loss by photoreduction indicate that this loss is consistent over time.

Because this manuscript has a focus on using Hg isotope ratios, this last sentence remains a bit unclear to me whether you mean an archive for atmospheric Hg deposition or also the atmospheric Hg isotope ratios (would the Hg isotope ratios also be conserved in the peat associated Hg?).

Many thanks for these comments! We have added the calculation method of photo-reductive Hg loss in supplementary Text S4. To be more precise, it is ca.10% of Hg loss via streamflow ($1.6 \mu\text{g m}^{-2} \text{yr}^{-1}$ in streamflow / $17.1 \mu\text{g m}^{-2} \text{yr}^{-1}$ total Hg deposition). Since boreal forest runoff Hg isotope composition is the same as the one in the soil (Jiskra et al., 2017)²⁸, we argue that the 10% loss of Hg via runoff does not alter the peat Hg isotope signatures and thus conserves the atmospheric Hg input signals. We agree that the role of lateral transport and evasion need to be included, but that this does not compromise our overall interpretation of the findings. The evasion from the vegetation at the surface is already mentioned in the second sentence of the Conclusion section, so we think it is just the lateral export that needs mentioned specifically. We have revised lines 380-388 accordingly:

“... This means we have not found any reduction mechanisms that would significantly redistribute Hg in gaseous form once it is incorporated into the peat profile and thereby change the peat Hg archive during decomposition processes. There is a small amount of the previously deposited Hg (10%) that is exported downstream together with dissolved organic matter in stream runoff. Peat soil Hg isotopes are likely not significantly being altered under stream runoff based on similar Hg isotope composition between boreal forest runoff and soils²⁷. The processes of Hg transformation we unravel here would constrain the mobility of Hg while the peat OM slowly decays below the groundwater table, and thus the possibilities for mobilization of Hg deposited from the atmosphere in earlier decades or centuries”.

We have also revised the sentence in lines 393-395 to make it more reasonable as follows:

“We do, however, suggest that the redox transformation processes during peat diagenesis are not significant enough to disqualify peat soil to be a reasonably reliable archive of long-term atmospheric Hg deposition patterns.”

Methods:

The presented data is of sufficient quality, despite the higher analytical uncertainty in the MIF data. The authors mostly used appropriate quality controls. However there are some questions regarding the methods that need clarification. Further, the calculations of accumulation rates or loss rates are not well explained and it is not entirely possible to follow all the steps or even replicate the calculations. The authors should provide a more detailed description of the calculation steps and the input parameters used.

The effort needed to obtain Hg isotope data from peat groundwater is remarkable. However, it should also be noted that the dataset is therefore also limited and that any interpretations should be made with caution.

Many thanks again for your recognition of our work and very helpful comments to improve our manuscript! Indeed, we discuss our datasets with caution to avoid over-interpretation and we are also honest with the limitations of the dataset. But we also believe that our study can contribute to the deep understanding of the post-depositional Hg transformation in peatlands.

We have added the information about the calculations of Hg accumulation rates and Hg loss in lines 527-529 and supplementary Text S4.

L 454: Why is cleaning of peat slices with MilliQ water necessary?

It is actually not 100% necessary. That we did this was to avoid any potential debris left from the cutting procedure if there is any.

XANES results are reported in the SI and only referred to in the main text without stating that these are the results of XANES analysis and without reference to the SI. This should be more clearly mentioned and referenced.

Thanks for this helpful comment! We have now made it clear about the objective of XANES analysis, by adding *“To obtain the information of sulfur species for examining the reduction/oxidation condition in peatland”* in lines 450-451. We have also mentioned the figures by referring to *“Supplementary figure S10, S11”* in lines 458-459.

The sampling and pre-concentration for the groundwater samples needs some additional explanations. In order to sample 2000 L of water this needed approx. 100 bottles? How were the samples stored in between? The pre-concentration needs some specification how this large volume was handled (Fig. S13).

We collected four to eight 20L bottles at a daily basis over the period of 23rd June to 13th July 2021. Once collected and shipped back to the lab, we immediately removed 4L of peat water to get 16L peat water, then started the pre-concentration. We did not store any peat water samples for DGM pre-concentration since storage. Relevant information has been added in the caption of Figure S14.

How was the recovery (yield) calculated? Was an aliquot of the peat groundwater sampled for isotope analysis also used to also analyze total Hg concentration? The methods only specify how DGM was measured in groundwater (Fig. S14). I assume this was the same setup for rainfall DGM analysis?

The recovery is calculated by the measured DGM concentration and total volume of peat water for pre-concentration relative to the analyzed Hg concentration using CV-AFS method.

An aliquot of the peat groundwater sampled for isotope analysis also used to measure DGM concentration.

Yes, it is the same setup for both DGM ground water and rainfall DGM. More relevant information has been added to the caption of supplementary Figure S15.

L 525-530: Rainfall DGM results should be reported in the SI along with the other results. Is there any control on how much Hg would be lost from purging a Hg(II) solution with similarly low concentrations? How were total Hg concentrations in rainfall determined?

Thanks for your comments! Rainfall DGM results are now reported in supplementary Table S7. To guarantee the reliable output of DGM analysis without any Hg loss or significant Hg transformation from Hg^{II} to Hg⁰, 1/we covered the sampling bottles with black plastic bags to prevent sunlight influence, 2/ we measured the samples immediately after the field trips, and 3/ we tested 0 ng Hg in the analytical line prior to any analysis. Unfortunately, we did not directly measure total Hg concentration in rainfall in 2022, but rainfall Hg concentration has been well reported in our colleague's work (Osterwalder et al., 2017)²⁹.

Regarding the analysis of rainfall Hg concentration, it is the same setup for both DGM ground water and rainfall DGM. Relevant information has been added to the caption of supplementary Figure S15.

L 533: I assume the four measurements are the reported DGM values. It does not become apparent in the main text that the results for groundwater DGM are aliquots of the same sampling. The effort of obtaining this data is huge and should be honored, but nonetheless this should be made transparent.

Thank you for pointing this out! Sorry that it was not clear enough in the original section. We aim for transparency in our methods and data interpretation. We have now added more relevant information in lines 502-504 to make it clear, *“Due to the low Hg⁰ concentration in peat groundwater, approximately 2000 L of peat water was collected for pre-concentration from 23rd June to 13th July 2021, enabling four aliquots of DGM isotope measurements (i.e. 10 ng Hg⁰ per measurement).”*.

L 535: Fig. S3 shows a calibrated age-depth model. You mean Fig. S13?

Yes, it should be the original Fig. S13. Thank you! It is now revised to Fig. S14 following the changes made in the supplementary figures.

L 536: Were multiple of the setups depicted in Fig. 13 run in parallel? In order to process 2000 L of water this probably needed several days of work? “Within 1h” becomes much less meaningful and I wonder how the remaining water was stored?

Yes, there were four setups running in parallel. It took us 20 field trips with four 20L bottles each from 23rd Jun to 13th July 2020. We did not store any peat water for DGM pre-concentration for isotope analysis since the storage overnight or longer might cause the transformation between DGM and Hg^{II}.

L 547: The “150ml rainfall and peat groundwater samples” are the ones in L 529-530? This is a bit confusing after the pre-concentration section describing other groundwater samples, plus the 150ml are not mentioned before and can't be used to identify which samples are meant.

Thanks a lot for this helpful comment to make our manuscript clear! Now the sentence has been rephrased to “*Once the rainfall and peat groundwater samples were collected, the DGM was immediately analyzed for concentration on a Tekran 2537X (supplementary Figure S15).*”. We have also added information about the volume of 150ml in the caption of supplementary Figure S15.

L 561: Were the Fluka standards matrix matched? 20% acid strength is rather on the upper limit for efficient reduction in the CV system.

Yes.

L 563: Standard-sample bracketing can be used to minimize effects of signal drifts during the measurement, however it cannot correct instrumental mass bias. Established methods for Hg isotope analysis using MC-ICP-MS usually involve both standard-sample bracketing and instrumental mass bias correction with thallium doping. The cited reference (Ref. 88) suggests: “We strongly suggest first correcting instrumental mass bias by simultaneous measurement of the certified thallium isotope standard (NIST SRM 997; 205Tl / 203Tl ratio of 2.38714) followed by sample-standard bracketing with the NIST SRM 3133 Hg standard.” Is there a reason there was no Tl doping used in this study?

Thank you for these comments! The reason that we did not use Tl doping was because Hg standard NIST 3133 bracketing is enough (Sun et al., 2013)³⁰.

The method section should also include details about how the different Rayleigh models displayed in the Figures 4 and used to calculate photoreductive Hg loss were implemented. The calculation of the reductive loss, especially for older periods is not explained and should definitely be added. Further, the calculation of HgAR is not explained and details should be added.

We have added the information about the calculations of Hg accumulation rates and Hg loss in lines 527-529 and supplementary Text S4, as responded earlier.

Supporting information:

Figure S1: d) In order to sample 2000 L of water multiple such bottles must have been sampled? This must have taken quite long. How were the individual bottles stored? Was there a difference in Hg concentration (or other parameters for that matter) between the initial sampling and the last sampling?

As responded earlier, It took us 20 field trips with four 20L bottles each from 23rd Jun to 13th July. We did not store any peat water for DGM pre-concentration for isotope analysis.

We kept track on the DGM concentration in peat water. The DGM concentration profile overall shows a decreasing trend as shown in the figure below.

Figure S11: see comment above, please add a description of the method approach used to calculate the loss rates.

We have added the information about the calculations of Hg accumulation rates and Hg loss in lines 527-529 and supplementary Text S4, as responded earlier.

Table S6: How was the yield calculated? "Hg recovered from trap and sampling yield" is not very clear. "Hg recovered from trap": Did you measure Hg(0) concentration before and then calculate the recovery after purging it onto the traps? What is meant with "sampling yield"?

Yes, we measured DGM concentration before and calculate the recovery after purging it onto the traps. The "Hg recovered from the trap" is calculated by the measured DGM concentration and total volume of peat water for pre-concentration. The "sample yield" means the ratio of analyzed Hg concentration using CV-AFS method and Hg recovered from the trap. To clarify these two methods, relevant information has been added in the footnote of Table S6.

Table S7: The 2SD for $\Delta^{201}\text{Hg}$ in Fig. S8 looks larger than the reported 0.16 ‰? Please verify that it is the same value.

The maximum reported value of $\Delta^{201}\text{Hg}$ is 0.19‰ (procedural standard NIST 1515), as shown in line 548 of the main text and in the supplementary Table S8. The value of 0.19‰, instead of 0.16‰, is presented in the supplementary Figure S9.

Table S2 and S5: There is a relationship between $\Delta^{201}\text{Hg}$ and $\Delta^{199}\text{Hg}$ and generally these two values should follow similar trends. However, depending on the process involved (MIE, NVE) the relationship can be different. What was the exact criterion for the exclusion of certain $\Delta^{201}\text{Hg}$ results? This explanation should be further clarified.

It was due to the high uncertainties of the $\Delta^{201}\text{Hg}$ results in the CRMs (up to 0.19‰), resulting from some unknown reasons in the multi-collector ICPMS.

References

1. Sklyberg, U., Bloom, P. R., Qian, J., Lin, C.-M. & Bleam, W. F. Complexation of Mercury(II) in Soil Organic Matter: EXAFS Evidence for Linear Two-Coordination with Reduced Sulfur Groups. *Environ. Sci. Technol.* **40**, 4174–4180 (2006).
2. Sklyberg, U. Competition among thiols and inorganic sulfides and polysulfides for Hg and MeHg in wetland soils and sediments under suboxic conditions: Illumination of controversies and implications for MeHg net production: COMPETITION AMONG S LIGANDS FOR Hg. *J. Geophys. Res.* **113**, n/a-n/a (2008).
3. Peichl, M. *et al.* Energy exchange and water budget partitioning in a boreal minerogenic mire: PEATLAND ENERGY AND WATER EXCHANGES. *J. Geophys. Res. Biogeosci.* **118**, 1–13 (2013).
4. Campeau, A. *et al.* Aquatic export of young dissolved and gaseous carbon from a pristine boreal fen: Implications for peat carbon stock stability. *Glob Change Biol* **23**, 5523–5536 (2017).
5. Bergquist, B. A. & Blum, J. D. Mass-Dependent and -Independent Fractionation of Hg Isotopes by Photoreduction in Aquatic Systems. *Science* **318**, 417–420 (2007).
6. Zheng, W. & Hintelmann, H. Mercury isotope fractionation during photoreduction in natural water is controlled by its Hg/DOC ratio. *Geochimica et Cosmochimica Acta* **73**, 6704–6715 (2009).
7. Zheng, W. & Hintelmann, H. Isotope Fractionation of Mercury during Its Photochemical Reduction by Low-Molecular-Weight Organic Compounds. *J. Phys. Chem. A* **114**, 4246–4253 (2010).
8. Bergman, I. *et al.* The Influence of Sulphate Deposition on the Seasonal Variation of Peat Pore Water Methyl Hg in a Boreal Mire. *PLoS ONE* **7**, e45547 (2012).
9. Fritsche, J. *et al.* Evasion of Elemental Mercury from a Boreal Peatland Suppressed by Long-Term Sulfate Addition. *Environ. Sci. Technol. Lett.* **1**, 421–425 (2014).
10. Åkerblom, S. *et al.* Formation and mobilization of methylmercury across natural and experimental sulfur deposition gradients. *Environmental Pollution* **263**, 114398 (2020).
11. Nijp, J. J. *et al.* Can frequent precipitation moderate the impact of drought on peatmoss carbon uptake in northern peatlands? *New Phytol* **203**, 70–80 (2014).
12. McCarter, C. P. R. & Price, J. S. Ecohydrology of *Sphagnum* moss hummocks: mechanisms of capitula water supply and simulated effects of evaporation: ECOHYDROLOGY OF SPHAGNUM MOSS HUMMOCKS. *Ecohydrol.* **7**, 33–44 (2014).
13. Nijp, J. J. *et al.* Including hydrological self-regulating processes in peatland models: Effects on peatmoss drought projections. *Science of The Total Environment* **580**, 1389–1400 (2017).
14. Enrico, M. *et al.* Atmospheric Mercury Transfer to Peat Bogs Dominated by Gaseous Elemental Mercury Dry Deposition. *Environ. Sci. Technol.* **50**, 2405–2412 (2016).
15. Gilli, R. *et al.* Speciation and Mobility of Mercury in Soils Contaminated by Legacy Emissions from a Chemical Factory in the Rhône Valley in Canton of Valais, Switzerland. *Soil Syst.* **2**, 44 (2018).
16. Jiskra, M. *et al.* Mercury Deposition and Re-emission Pathways in Boreal Forest Soils Investigated with Hg Isotope Signatures. *Environ. Sci. Technol.* **49**, 7188–7196 (2015).
17. Zheng, W. *et al.* Mercury Stable Isotope Fractionation during Abiotic Dark Oxidation in the Presence of Thiols and Natural Organic Matter. *Environ. Sci. Technol.* **53**, 1853–1862 (2019).
18. Zheng, W., Foucher, D. & Hintelmann, H. Mercury isotope fractionation during volatilization of Hg(0) from solution into the gas phase. *J. Anal. At. Spectrom.* **22**, 1097 (2007).
19. Yuan, W. *et al.* Stable Isotope Evidence Shows Re-emission of Elemental Mercury Vapor Occurring after Reductive Loss from Foliage. *Environ. Sci. Technol.* **53**, 651–660 (2019).
20. Sun, R. *et al.* Historical (1850–2010) mercury stable isotope inventory from anthropogenic sources to the atmosphere. *Elementa: Science of the Anthropocene* **4**, 000091 (2016).
21. Lee, J. H. *et al.* Spatiotemporal Characterization of Mercury Isotope Baselines and Anthropogenic Influences in Lake Sediment Cores. *Global Biogeochem Cycles* **35**, (2021).
22. Yin, R., Lepak, R. F., Krabbenhoft, D. P. & Hurley, J. P. Sedimentary records of mercury stable isotopes in Lake Michigan. *Elementa: Science of the Anthropocene* **4**, 000086 (2016).

23. Enrico, M. *et al.* Holocene Atmospheric Mercury Levels Reconstructed from Peat Bog Mercury Stable Isotopes. *Environ. Sci. Technol.* **51**, 5899–5906 (2017).
24. Kurz, A. Y., Blum, J. D., Washburn, S. J. & Baskaran, M. Changes in the mercury isotopic composition of sediments from a remote alpine lake in Wyoming, USA. *Science of The Total Environment* **669**, 973–982 (2019).
25. Zerkle, A. L. *et al.* Anomalous fractionation of mercury isotopes in the Late Archean atmosphere. *Nat Commun* **11**, 1709 (2020).
26. Jiskra, M. *et al.* Climatic Controls on a Holocene Mercury Stable Isotope Sediment Record of Lake Titicaca. *ACS Earth Space Chem.* **6**, 346–357 (2022).
27. Li, C. *et al.* A peat core Hg stable isotope reconstruction of Holocene atmospheric Hg deposition at Amsterdam Island (37.8oS). *Geochimica et Cosmochimica Acta* **341**, 62–74 (2023).
28. Jiskra, M., Wiederhold, J. G., Skjellberg, U., Kronberg, R.-M. & Kretzschmar, R. Source tracing of natural organic matter bound mercury in boreal forest runoff with mercury stable isotopes. *Environ. Sci.: Processes Impacts* **19**, 1235–1248 (2017).
29. Osterwalder, S. *et al.* Mercury evasion from a boreal peatland shortens the timeline for recovery from legacy pollution. *Sci Rep* **7**, 16022 (2017).
30. Sun, R., Enrico, M., Heimbürger, L.-E., Scott, C. & Sonke, J. E. A double-stage tube furnace—acid-trapping protocol for the pre-concentration of mercury from solid samples for isotopic analysis. *Anal Bioanal Chem* **405**, 6771–6781 (2013).

Reviewer #1 (Remarks to the Author):

I have reviewed the revised manuscript and the responses by the authors to my previous comments. I am totally satisfied by the responses and I have no further hold back for this manuscript to be published besides other potential reviewers' comments. Thank you for the great work on how we can better use Hg isotopes for these field data interpretation.

Reviewer #2 (Remarks to the Author):

I have reviewed the authors' responses to my first-round comments as well as the changes they have made to the manuscript. I am satisfied with the responses and changes made. Well done.

Reviewer #3 (Remarks to the Author):

I would like to thank the authors for the very detailed answers and explanations to the many questions and comments, the careful consideration is appreciated. Overall, I believe that with all the clarifications and added information the manuscript is more robust and there are no further clarifications or changes required from my side. Again, I'd like to point out the great effort needed to obtain such data and highlight the value of such studies and support the publication of this manuscript in Nature Communications.

The only point I do not fully agree is the use of TI doping for mass bias correction during Hg isotope analysis, even though the authors provide an example where it made no difference. When reporting a dataset in which one of the isotopes cannot be used because of unknown errors during the measurement, arguing that a common QC method is not necessary is not most convincing. In my experience Hg isotope analyses are more reliable with TI doping even under relatively stable instrument conditions. Plus, the effort of obtaining and preparing field samples like these for isotope analysis is in no relation to the small effort of using TI doping for mass bias correction. However, I also fully understand that this cannot be changed and is more something to consider for future work.

We thank very much the reviewers for their recognition of our work and further comments! Please see below our point-by-point response in blue.

Reviewer #1 (Remarks to the Author):

I have reviewed the revised manuscript and the responses by the authors to my previous comments. I am totally satisfied by the responses and I have no further hold back for this manuscript to be published besides other potential reviewers' comments. Thank you for the great work on how we can better use Hg isotopes for these field data interpretation.

We feel grateful to the reviewer for his/her recognition of our work!

Reviewer #2 (Remarks to the Author):

I have reviewed the authors' responses to my first-round comments as well as the changes they have made to the manuscript. I am satisfied with the responses and changes made. Well done.

We appreciate very much the first-round comments made by the reviewer!

Reviewer #3 (Remarks to the Author):

I would like to thank the authors for the very detailed answers and explanations to the many questions and comments, the careful consideration is appreciated. Overall, I believe that with all the clarifications and added information the manuscript is more robust and there are no further clarifications or changes required from my side. Again, I'd like to point out the great effort needed to obtain such data and highlight the value of such studies and support the publication of this manuscript in Nature Communications.

We feel grateful to the reviewer for his/her recognition of our work!

The only point I do not fully agree is the use of Tl doping for mass bias correction during Hg isotope analysis, even though the authors provide an example where it made no difference. When reporting a dataset in which one of the isotopes cannot be used because of unknown errors during the measurement, arguing that a common QC method is not necessary is not most convincing. In my experience Hg isotope analyses are more reliable with Tl doping even under relatively stable instrument conditions. Plus, the effort of obtaining and preparing field samples like these for isotope analysis is in no relation to the small effort of using Tl doping for mass bias correction. However, I also fully understand that this cannot be changed and is more something to consider for future work.

Thanks a lot for your further comments! We agree that the use of Tl doping for mass bias correction during Hg isotope analysis is rather reliable. But our work analyzed on Nu (ETH Zurich) is also trustworthy based on the good data quality on reference materials (ETH_Fluka and NIST SRM 1515 in supplementary Table S8). In addition, some other reliable analytical work without Tl doping are also reported (e.g., Jiskra et al., 2021_Nature, <https://doi.org/10.1038/s41586-021-03859-8>). However, we think that it would be interesting to make a comparison on Hg isotope analysis in peat samples with and without Tl doping. We intend to do it in the future.